# Sequential Kernel-based Conditional Independence Testing via Adaptive Betting

**Zheng He** [1]   **Danica J. Sutherland** [1] [2]

## Abstract

Testing conditional independence is fundamental yet intrinsically difficult: without additional assumptions, Type I error control is impossible in general. The "Model-X" paradigm addresses this difficulty by assuming exact knowledge of a relevant conditional distribution. While small deviations from this assumption can sometimes be tolerated in classical one-shot testing, existing sequential conditional independence tests typically require the Model-X conditional to be known exactly, making them fragile when it must instead be estimated. We propose a new approach that is substantially more robust to such estimation error. Our method applies testing-by-betting to an adaptively optimized Kernel Conditional Independence statistic, together with a normalization scheme and a truncate-and-shift calibration strategy. These modifications greatly reduce Type I error inflation while preserving high power across high-dimensional synthetic benchmarks and real-world fairness tasks, outperforming existing sequential Model-X approaches. Code is available at https://github.com/he-zh/SKCI.

## 1. Introduction

Do customers pay more for car insurance based on their race, even after controlling for neighborhood risk factors (Angwin et al., 2017)? Can predictive models satisfy the widely used notion of *equalized odds* fairness, thereby mitigating such disparities (Hardt et al., 2016)? Does a pedestrian detection model remain robust across environments and weather conditions, even when the true distribution of pedestrians shifts (Jiang & Veitch, 2022)? Are the shapes of a patient's cancer cells associated with the dosage of a particular medication, after controlling for demographics

and disease progression (Zhao et al., 2026)? These, and many more, fundamental questions can all be formulated as problems of measuring or testing *conditional independence* (CI): given sample triples $(A, B, C)$, we seek to reject the null hypothesis that $A \perp\!\!\!\perp B \mid C$.

Classical approaches to null hypothesis testing, however, have come under increasing criticism. Widespread concerns about the "reproducibility crisis" in empirical science have highlighted the fragility of classical p-value-based inference under flexible data collection, multiple testing, and optional stopping (e.g. Gelman & Loken, 2013). In complex scientific and machine learning workflows, where data often arrive sequentially and analyses may adapt over time, more flexible tools are needed. The theory of anytime-valid testing based on e-values and testing by betting (see, e.g. Ramdas & Wang, 2025) provides a principled alternative while remaining compatible with likelihood-based reasoning and machine-learning-based estimators.

In this paper, we focus on sequential testing with an i.i.d. data stream, where obbservations arrive sequentially and the test may be stopped at a data-dependent time. Prior anytime-valid CI tests (Shaer et al., 2023; Pandeva et al., 2024a;b) operate in the Model-X setting (Candès et al., 2018), assuming that the conditional law $P_{A|C}$ is known exactly. This assumption is powerful because it avoids the usual impossibility results for CI testing: without further assumptions, no test can achieve nontrivial power in either the fixed-sample (Shah & Peters, 2020) or online setting (Waudby-Smith & Ramdas, 2023). We discuss these hardness results in Section 2.3.

Exact knowledge of $P_{A|C}$, however, is often unrealistic – especially in online settings. We therefore propose a sequential test for CI that works well when this conditional law is known, but is designed to remain effective when the relevant conditional distributions must be estimated online. Since exact false-rejection control is not possible in full generality without additional assumptions, we provide theoretical and empirical evidence that the method can retain good Type I error control and reasonable power in this estimated-conditional regime.

In developing this test, we also formalize a principle for choosing test statistics that accumulate evidence quickly under weak signal. This principle is not specific to CI testing

[1]Department of Computer Science, University of British Columbia, Vancouver, Canada [2]Alberta Machine Intelligence Institute, Edmonton, Canada. Correspondence to: Zheng He <zhhe@cs.ubc.ca>, Danica J. Sutherland <dsuth@cs.ubc.ca>.

*Proceedings of the 43rd International Conference on Machine Learning*, Seoul, South Korea. PMLR 306, 2026. Copyright 2026 by the author(s).

and may be useful more broadly for testing by betting.

## 2. Background

We begin by establishing the formal framework for sequential hypothesis testing through the betting paradigm, which reformulates statistical testing as an online game.

### 2.1. Sequential Hypothesis Testing via Betting

Consider a stream of i.i.d. observations $(Z_t)_{t \geq 1}$, where each $Z_t \in \mathcal{Z}$ arrives sequentially. Our goal is to test a null hypothesis $H_0$ regarding the distribution of this sequence. Unlike fixed-sample tests, a sequential test may reject at any data-dependent time. Its Type I error therefore measures the probability of ever rejecting $H_0$ under the null.

The framework of *testing by betting* (overviewed by Ramdas & Wang, 2025) reinterprets hypothesis testing as a game where a player attempts to accumulate wealth by betting against the null. The game is constructed such that no strategy can systematically profit if $H_0$ is true; conversely, a successful betting strategy under the alternative $H_1$ yields capital growth, making the accumulated wealth a direct and intuitive measure of evidence against the null.

A player starts with an initial wealth $W_0 = 1$. At each round $t$, before observing $Z_t$, the player chooses a *payoff function* $f_t : \mathcal{Z} \to [-1, \infty)$ designed to capture the discrepancy between $H_0$ and $H_1$, and bets a fraction $\lambda_t \in [0, 1]$ of their wealth; both $f_t$ and $\lambda_t$ should be $\mathcal{F}_{t-1}$-measurable, where $\mathcal{F}_{t-1} = \sigma(Z_1, \ldots, Z_{t-1})$ denotes the history observed before round $t$. Upon observing $Z_t$, their wealth becomes

$$W_t = W_{t-1}(1 + \lambda_t f_t(Z_t)).$$

The test will be valid if $\mathbb{E}_{H_0}[f_t(Z_t) \mid \mathcal{F}_{t-1}] \leq 0$, as this makes the wealth process $(W_t)_{t \geq 0}$ a non-negative *supermartingale* under $H_0$: $\mathbb{E}_{H_0}[W_t \mid \mathcal{F}_{t-1}] \leq W_{t-1}$. This allows for the application of Ville's inequality (1939), showing that for any $\alpha \in (0, 1)$,

$$\Pr_{H_0}(\exists t \geq 1 : W_t \geq 1/\alpha) \leq \alpha \, \mathbb{E}_{H_0}[W_0] = \alpha.$$

Thus we construct an *anytime valid* test of level $\alpha$ by rejecting the null at the stopping time $\tau = \inf\{t \geq 1 : W_t \geq 1/\alpha\}$. Among valid tests, we would like ones which choose $f_t$ and $\lambda_t$ so that $W_t$ grows rapidly under the alternative $H_1$, e.g. by maximizing the e-power $\mathbb{E}_{H_1}[\log W_t]$.

### 2.2. Designing Payoff Functions

In most existing literature, $f_t$ is constructed by contrasting a statistic $g_t(\cdot)$ evaluated on the observed data $Z_t$ against the same statistic evaluated on a null-calibrated sample $\widetilde{Z}_t$,

$$f_t(Z_t) = g_t(Z_t) - g_t(\widetilde{Z}_t),$$

where $\widetilde{Z}_t$ is constructed from $Z_t$ in a way such that $\mathbb{E}[g_t(\widetilde{Z}_t) \mid \mathcal{F}_{t-1}] = \mathbb{E}_{H_0}[g_t(Z_t) \mid \mathcal{F}_{t-1}]$.

For instance, in *two-sample testing*, $Z_t = (A_t, B_t)$ for stationary and independent $A$ and $B$; we wish to test the null hypothesis that the distribution of $A$ equals that of $B$. Then we might use $\widetilde{Z}_t = (B_t, A_t)$, since under $H_0$ the two distributions are the same (Shekhar & Ramdas, 2023).

We would like to select $g$ to distinguish the distributions as confidently as possible under $H_1$, e.g. by choosing a function class $\mathcal{G}$ and attempting to pick

$$\mathbb{E}[f^*(Z_t, \widetilde{Z}_t)] = \sup_{g \in \mathcal{G}} \Big( \mathbb{E}[g(Z_t)] - \mathbb{E}[g(\widetilde{Z}_t)] \Big). \quad (1)$$

If $-g \in \mathcal{G}$ for all $g \in \mathcal{G}$, the right-hand side is an integral probability metric (Müller, 1997), a common way to quantify distributional discrepancies.

**Current Paradigms and Limitations** Shekhar & Ramdas (2023) and Podkopaev et al. (2023) restrict $\mathcal{G}$ to the unit ball of a Reproducing Kernel Hilbert Space (RKHS), where (1) becomes the Maximum Mean Discrepancy (MMD; Gretton et al., 2012) between the distributions of $Z_t$ and $\widetilde{Z}_t$. Shaer et al. (2023) use the squared prediction error of a model trained on past data as one example of $g$. DAVT (Pandeva et al., 2024b) instead chooses a neural network class for $\mathcal{G}$.

In all of these cases, the power depends on both the underlying distributional discrepancy and the quality of the chosen statistic $g$. If the discrepancy is small, even a well-chosen statistic leads to slow wealth growth. This problem cannot be dodged by arbitrarily scaling $\mathcal{G}$, since level control via Ville's inequality requires a nonnegative wealth process and hence $f_t(Z_t, \widetilde{Z}_t) \geq -1$ almost surely.

### 2.3. Challenges in Null Calibration for CI Testing

For two-sample testing or unconditional independence testing ($A \perp\!\!\!\perp B$), exact null-calibrated samples $\widetilde{Z}$ can be easily obtained via permutation of the samples. For conditional independence (CI) testing, where we ask if $A \perp\!\!\!\perp B \mid C$, generating $\widetilde{Z}$ is substantially more difficult. This is especially true if $C$ is continuous, in which case we would typically observe only a single $(A, B)$ pair for any given value of $C$.

Shah & Peters (2020) prove that no test with exact Type I error control for all continuous null distributions can have nontrivial power. For any testing procedure, any sample size, and any Lebesgue-continuous joint distribution of $(A, B, C)$ with $A \not\perp\!\!\!\perp B \mid C$, there exists another continuous joint distribution $(A', B', C')$ satisfying $A' \perp\!\!\!\perp B' \mid C'$ that is indistinguishable to the test at this number of samples, even if all distributions are restricted to have bounded support. (The dependence can be hidden in "hard-to-find" features of the conditioning variable $C$, such as lower-order bits.)

Waudby-Smith & Ramdas (2023) extend this hardness phenomenon to sequential testing, showing that anytime-valid CI tests cannot guarantee nontrivial power uniformly over all alternatives, even if the test is forced to wait for an arbitrarily large number of observations before stopping.[1]

To avoid these impossibility results, much of the current CI testing literature imposes additional structure. The most common example is the *Model-X assumption*, which assumes the conditional distribution $P_{A|C}$ is known (Candès et al., 2018; Shaer et al., 2023; Pandeva et al., 2024b; Grünwald et al., 2024). Under this assumption, one can sample $\widetilde{A} \sim P_{A|C}$ and form $\widetilde{Z} = (\widetilde{A}, B, C)$, which follows the null distribution by construction. Other methods instead assume that $P_{A,C}$ is invariant across the null and alternative (Pandeva et al., 2024a). With such additional structure, exactly valid CI tests can be constructed.

In practice, we rarely have *perfect* knowledge of the distribution of $A \mid C$. One might instead estimate this conditional distribution from an auxiliary sample of $(A, C)$ pairs, so that $\widetilde{Z}$ only approximately follows the null. A sufficiently powerful test – whether because it has many test observations or because it identifies a strong distinguishing statistic $g$ – may then detect the mismatch between $Z$ and the incorrectly generated $\widetilde{Z}$, and reject even when $A \perp\!\!\!\perp B \mid C$. Pogodin et al. (2024) and He et al. (2025) study related calibration failures for conditional testing in the batch setting. The online setting is even more demanding, since validity must hold across an unbounded number of stopping opportunities, and approximation error in $\widetilde{Z}$ must remain small enough to not become detectable as arbitrarily more observations arrive. Moreover, the auxiliary-data regime itself can be unrealistic in applications where $(A, B, C)$ triples arrive only online.

In this work, we avoid constructing an explicit null-calibrated sample $\widetilde{Z}$. Instead, we use payoffs of the form $f_t(Z_t) = g_t(Z_t; \gamma_t)$, where $\gamma_t$ is a data-dependent shift chosen to make the resulting wealth process approximately a supermartingale under the null.

## 3. Methodology

In this section, we present our sequential testing framework for conditional independence. We construct betting payoffs that are designed to promote rapid wealth growth under alternatives while mitigating Type I error inflation beyond the exact Model-X setting.

Consider a stream of i.i.d. observations $Z_t = (A_t, B_t, C_t)$, each in a space $\mathcal{Z} = \mathcal{A} \times \mathcal{B} \times \mathcal{C}$, arriving in batches. To ensure that the payoff function and the betting fraction are predictable ($\mathcal{F}_{t-1}$-measurable), we adopt a sequential data partitioning scheme that separates model estimation, calibration, and testing. At each round t, the observed data are divided into three disjoint subsets:

- **Training set** ($\mathcal{X}_{t-1}^{\text{tr}}$): Used to estimate the data-dependent quantities appearing in the test statistic. This set grows monotonically over time.

- **Validation set** ($\mathcal{X}_{t-1}^{\text{val}}$): A held-out buffer used to determine calibration quantities required by the test.

- **Test batch** ($\mathcal{Y}_t$): Fresh observations used to update the wealth process $W_t$.

Let $\mathcal{H}_{t-1} = \mathcal{X}_{t-1}^{\text{tr}} \cup \mathcal{X}_{t-1}^{\text{val}}$ denote the historical data available prior to observing $\mathcal{Y}_t$, and define the associated filtration $\mathcal{F}_{t-1} = \sigma(\mathcal{H}_{t-1})$. We update the wealth according to

$$W_t = W_{t-1}(1 + \lambda_t V_t), \quad \lambda_t \in (0,1), \qquad (2)$$

where $V_t$ denotes the payoff used at round $t$, corresponding to $f_t(\mathcal{Y}_t)$ in the notion of payoff functions from Section 2. If $\lambda_t$ is $\mathcal{F}_{t-1}$-measurable, and $\mathbb{E}_{H_0}[V_t \mid \mathcal{F}_{t-1}] \leq 0$, then the wealth process is a supermartingale under $H_0$.

Before the next batch, we update the data partitions as

$$\mathcal{X}_t^{\text{tr}} \leftarrow \mathcal{X}_{t-1}^{\text{tr}} \cup \mathcal{X}_{t-1}^{\text{val}}, \qquad \mathcal{X}_t^{\text{val}} \leftarrow \mathcal{Y}_{t-1}.$$

The batch of data observed at timestep $t$ is used first as $\mathcal{Y}_t$, then as $\mathcal{X}_{t+1}^{\text{val}}$, and then for all $t' \geq t+2$ is included in $\mathcal{X}_{t'}^{\text{tr}}$.

**A Self-Normalized Payoff** We will build our measure of discrepancy between $H_0$ and $H_1$ from a symmetric kernel[2] $h : \mathcal{Z} \times \mathcal{Z} \to \mathbb{R}$ designed to capture conditional independence. Ideally, $h$ vanishes on average under $H_0$, $\mathbb{E}_{H_0}[h(Z, Z') \mid Z] = 0$, and takes large values under $H_1$.

We then construct our "raw" betting score by comparing the new data to historical observations with $h$, using a form that can be described as a cross U-statistic (Kim & Ramdas, 2024). Given a training history $\mathcal{X}_{t-1}^{\text{tr}} = \{x_i\}_{i=1}^n$ and a fresh batch $\mathcal{Y}_t = \{y_j\}_{j=1}^b$, let

$$U_{n,b}(\mathcal{X}_{t-1}^{\text{tr}}, \mathcal{Y}_t) := \frac{1}{nb} \sum_{i=1}^n \sum_{j=1}^b h(x_i, y_j).$$

The "cross" structure is particularly convenient for taking conditional expectations. Conditional on the past data $\mathcal{F}_{t-1}$,

---

[1]It is worth briefly noting that these hardness results do not rule out *non-uniform* asymptotic control. For instance, the histogram-type CI test of Györfi & Walk (2012) is "strongly consistent" in a distribution-free sense, meaning that if the test is run with increasing sample sizes for any fixed distribution, it almost surely makes only finitely many mistakes. This number of mistakes will depend on the distribution, however, and so one cannot "wrap" this test into a procedure with distribution-free guarantees.

[2]"Kernel" is an extremely overloaded word in machine learning and statistics. We use it here in the sense of a $U$-statistic kernel, for which the only requirement is symmetry $h(z, z') = h(z', z)$. Our choice will also turn out to be the kernel of a reproducing kernel Hilbert space, but that is not necessary for our normalization.

the reference points $\mathcal{X}_{t-1}^{\mathrm{tr}} = \{x_i\}_{i=1}^n$ are fixed, while the new observation $\mathcal{Y}_t$ is independent of $\mathcal{F}_{t-1}$. Hence

$$\mathbb{E}\left[U_{n,b}(\mathcal{X}_{t-1}^{\mathrm{tr}}, \mathcal{Y}_t) \mid \mathcal{F}_{t-1}\right] = \frac{1}{n}\sum_{i=1}^n \mathbb{E}[h(x_i, Z) \mid F_{t-1}],$$

where $Z$ denotes an independent draw from the data-generating distribution. Under $H_0$, this conditional mean is ideally zero regardless of $x_i$.

When the discrepancy between $H_0$ and $H_1$ is weak (or $h$ is poor), however, the magnitude of $U_{n,b}$ can be small, leading to slow wealth growth under the alternative. To adapt to the unknown scale of the kernel interaction, we introduce the statistic $S_n$, computed entirely from historical data:

$$S_n(\mathcal{X}_{t-1}^{\mathrm{tr}}) := \frac{1}{n^2}\sum_{i=1}^n \sum_{j=1}^n h(x_i, x_j).$$

The V-statistic $S_n$ is $\mathcal{F}_{t-1}$-measurable, and so we can easily incorporate it into our bets. We use

$$V_t^{\mathrm{raw}} := \frac{U_{n,b}(\mathcal{X}_{t-1}^{\mathrm{tr}}, \mathcal{Y}_t)}{S_n(\mathcal{X}_{t-1}^{\mathrm{tr}}) + \varepsilon}, \quad \varepsilon > 0. \tag{3}$$

With large $n$ and $b$, both $U_{n,b}$ and $S_n$ converge to $\mathbb{E}\, h(X, Y)$. Thus, for small $\varepsilon$, $V_t^{\mathrm{raw}} \approx 1$ under alternative distributions, independently of the scale of $h$. Moreover, if $\mathbb{E}_{H_0}[h(x_i, Z) \mid \mathcal{F}_{t-1}] = 0$, then the numerator is conditionally mean zero under the null. Since the denominator is $\mathcal{F}_{t-1}$-measurable, the normalized $V_t^{\mathrm{raw}}$ remains conditionally mean zero, and the resulting wealth process is a martingale under the null. The regularization parameter $\varepsilon$ prevents instability when the denominator is close to zero, but should be much smaller than $\mathbb{E}_{H_1}\, h(X, Y)$ to preserve power.

**The KCI Operator**    We now specify our choice of $h$ via the Kernel-based Conditional Independence (KCI) framework (Zhang et al., 2011), which provides a principled RKHS representation of conditional independence.

Map $\mathcal{A}$, $\mathcal{B}$, and $\mathcal{C}$ into reproducing kernel Hilbert spaces $\mathcal{H}_A$, $\mathcal{H}_B$, and $\mathcal{H}_C$ with feature maps $\phi_A$, $\phi_B$, and $\phi_C$, respectively. Throughout, we assume that all RKHSs considered are separable and that the feature maps are measurable. The conditional mean embeddings $\mu_{A|C}(c) := \mathbb{E}[\phi_A(A) \mid C = c]$ and $\mu_{B|C}(c) := \mathbb{E}[\phi_B(B) \mid C = c]$ represent the components of $A$ and $B$ explained by $C$.

Following He et al. (2025), we construct the KCI operator $\psi(Z)$, which captures the residual interaction between $A$ and $B$ after conditioning on $C$:

$$\psi(z) = \big(\phi_A(a) - \mu_{A|C}(c)\big) \otimes \big(\phi_B(b) - \mu_{B|C}(c)\big) \otimes \phi_C(c).$$

Intuitively, $\psi(Z)$ encodes the dependence between the residuals of $A$ and $B$ after removing the effect of $C$, weighted

by the representation of $C$. Under the null, the residualized features are conditionally uncorrelated given $C$, and hence $\mathbb{E}_{H_0}[\psi(Z)] = 0$. Under suitable universality conditions on the kernels, $\mathbb{E}[\psi(Z)] \neq 0$ for any violation of conditional independence. We therefore define the kernel by

$$h(Z, Z') := \langle \psi(Z), \psi(Z') \rangle. \tag{4}$$

Then, for any fixed $Z$,

$$\mathbb{E}_{H_0}[h(Z, Z') \mid Z] = \langle \psi(Z), \mathbb{E}_{H_0}[\psi(Z')] \rangle = 0.$$

So far, we have treated the component kernels as fixed in advance and the conditional mean embeddings as known population quantities. In a practical sequential setting, however, the kernel may be updated as more data become available. Outside the Model-X setting, the conditional mean embeddings $\mu_{A|C}$ and $\mu_{B|C}$ are unknown and must be estimated from historical data $\mathcal{X}^{\mathrm{tr}}$.

At each round $t$, we construct the kernel $h^{(t)}$ using conditional mean embeddings estimated from the historical training data $\mathcal{X}_{t-1}^{\mathrm{tr}}$. Consequently, $h^{(t)}$ is allowed to depend on all past information, but remains $\mathcal{F}_{t-1}$-measurable. The round-$t$ payoffs $V^{\mathrm{raw}}$, and $V_t$ to be described next, are then computed from $(\mathcal{H}_{t-1}, \mathcal{Y}_t)$ using this kernel.

**The Shift-and-Truncate Mechanism**    To apply Ville's inequality, the betting payoff must satisfy two properties: $V_t \geq -1$ almost surely and $\mathbb{E}_{H_0}[V_t \mid \mathcal{F}_{t-1}] \leq 0$ for every round $t$. Although the kernel $h$ is bounded, the raw self-normalized score $V_t^{\mathrm{raw}}$ defined in (3) need not be bounded below by $-1$. In particular, fluctuations in the normalization term can produce large negative values, which would make the corresponding wealth update invalid.

We therefore define the payoff by applying a predictable shift $\gamma_t$ followed by a one-sided truncation:

$$V_t := \max\{V_t^{\mathrm{raw}} - \gamma_t, -1\}. \tag{5}$$

The truncation enforces $V_t \geq -1$, ensuring that the wealth process remains nonnegative. However, truncation alone can increase the conditional mean of the payoff. To preserve null calibration, we choose the smallest nonnegative shift $\gamma_t$ such that

$$\gamma_t := \min_{\gamma \geq 0}\Big\{\gamma : \mathbb{E}_{H_0}[\max\{V_t^{\mathrm{raw}} - \gamma, -1\} \mid \mathcal{F}_{t-1}] \leq 0\Big\}. \tag{6}$$

This defines an ideal predictable shift whenever the corresponding null expectations can be evaluated exactly.

Under $H_1$, when the kernel is aligned with the conditional dependence signal, the raw score $V_t^{\mathrm{raw}}$ has positive conditional mean and is less likely to fall far below zero. In this regime, the truncation is rarely active and the required shift $\gamma_t$ is relatively small.

Thus, the mechanism is designed to retain the power benefits of the self-normalized payoff. Exact anytime-valid Type I error control, however, is guaranteed for the ideal version in which the conditional null expectation in (6) is computed exactly. In practice, we approximate the expectation using the scheme described below.

**Gaussian Approximation for Shift Estimation**  The ideal shift in (6) depends on the conditional null distribution of the raw score $V_t^{\mathrm{raw}}$. Since this distribution is not available in general, we approximate it by a Gaussian law.

This approximation is motivated by the structure of $V_t^{\mathrm{raw}}$. For the round-$t$ test batch $\mathcal{Y}_t = \{y_j\}_{j=1}^b$, define

$$g^{(t)}(y_j) := \frac{1}{n\bigl(S_n(\mathcal{X}_{t-1}^{\mathrm{tr}}) + \varepsilon\bigr)} \sum_{i=1}^n h^{(t)}(x_i, y_j),$$

where $\{x_i\}_{i=1}^n \subseteq \mathcal{X}_{t-1}^{\mathrm{tr}}$. The raw score can be written as

$$V_t^{\mathrm{raw}} = \frac{1}{b} \sum_{j=1}^b g^{(t)}(y_j).$$

Conditional on the past information $\mathcal{F}_{t-1}$, each of the kernel $h^{(t)}$, the training points $\{x_i\}_{i=1}^n$, and the normalization $S_n(\mathcal{X}_{t-1}^{\mathrm{tr}}) + \varepsilon$ are fixed. Hence, $g^{(t)}$ is $\mathcal{F}_{t-1}$-measurable, and $V_t^{\mathrm{raw}}$ is a normalized average of test-sample contributions. As the test samples in the batch are independent, if $b$ is sufficiently large, by the central limit theorem

$$\mathrm{Law}\bigl(V_t^{\mathrm{raw}} \mid \mathcal{F}_{t-1}\bigr) \approx \mathcal{N}(\mu_t, \sigma_t^2).$$

We then approximate the shift by replacing the unknown conditional null law of $V_t^{\mathrm{raw}}$ in (6) with this Gaussian law. Under this approximation, the conditional null expectation in (6) can be evaluated in closed form. For a random variable $V \sim \mathcal{N}(\mu, \sigma^2)$, define

$$f(\gamma; \mu, \sigma) := \mathbb{E}_{V \sim \mathcal{N}(\mu, \sigma^2)}\bigl[\max\{V - \gamma, -1\}\bigr].$$

Defining $\xi := \frac{\gamma - \mu - 1}{\sigma}$, we can directly calculate that

$$f(\gamma; \mu, \sigma) = \sigma\left[\phi(\xi) - \xi\Phi(-\xi)\right] - 1,$$

where $\phi$ denotes the density of a standard normal and $\Phi$ its cumulative distribution function. The Gaussian plug-in shift is then chosen as the smallest nonnegative value of $\gamma$ for which $f(\gamma; \mu_t, \sigma_t) \leq 0$.

It remains to specify the plug-in parameters $\mu_t$ and $\sigma_t^2$. If the conditional mean embeddings used in $h^{(t)}$ are exact, then the raw score is centered under $H_0$. With estimated conditional mean embeddings, this centering is only approximate, but it is difficult to accurately estimate the true null mean; we therefore use the approximation $\widehat{\mu}_t = 0$.

Under the approximation $\widehat{\mu}_t = 0$, the conditional variance of of the normalized raw score satisfies

$$\mathrm{Var}_{H_0}[V_t^{\mathrm{raw}} \mid \mathcal{F}_{t-1}] = \frac{1}{b} \mathrm{Var}_{H_0}\left[g^{(t)}(Y) \mid \mathcal{F}_{t-1}\right]$$
$$\approx \frac{1}{b} \mathbb{E}_{H_0}[(g^{(t)}(Y))^2 \mid \mathcal{F}_{t-1}],$$

where $Y$ denotes an independent null sample.

During shift estimation, after these quantities have been fixed, we estimate null scale using the validation set $\mathcal{X}_{t-1}^{\mathrm{val}}$ and use this estimate to calibrate the final shift $\widehat{\gamma}_t$.

Let $\{v_j\}_{j=1}^b \subseteq \mathcal{X}_{t-1}^{\mathrm{val}}$ denote the validation points. We define

$$\widehat{\sigma}_t^2 := \frac{1}{b^2} \sum_{j=1}^b \bigl(g^{(t)}(v_j)\bigr)^2. \tag{7}$$

This separation reduces the bias that would arise from optimizing the score and estimating its null scale on the same training samples. The practical shift is then chosen as

$$\widehat{\gamma}_t := \min\Bigl\{\gamma \geq 0 : f(\gamma; 0, \widehat{\sigma}_t) \leq 0\Bigr\}. \tag{8}$$

Since $f(\gamma; 0, \widehat{\sigma}_t)$ is monotone nonincreasing in $\gamma$, $\widehat{\gamma}_t$ can be found efficiently by binary search. Restricting to $\gamma \geq 0$ is not strictly necessary, but makes the correction more conservative.

**Parameter Selection**  The kernel choices of KCI play two distinct roles in our procedure. First, the regression kernels used for conditional mean embedding estimation should be chosen to obtain good CME estimates. The kernel on the conditioning variable $C$ also affects the sensitivity of the KCI statistic to conditional dependence; this is particularly important in difficult CI testing problems where the signal can be hidden by an inappropriate conditioning kernel (He et al., 2025).

For the regression step, we follow Pogodin et al. (2024). We use fixed kernels for $A$ and $B$, with heuristic bandwidth choices, and select the regression kernels $k_{C \to A}$ and $k_{C \to B}$ by minimizing leave-one-out prediction error. These choices are used to estimate the conditional mean embeddings. Implementation details are given in Appendix B.2.

We then choose the conditioning kernel $k_C$ together with the betting fraction $\lambda_t$, since both affect the growth of the wealth process. In principle, the natural objective is the expected logarithmic wealth increment,

$$\arg\max_{\lambda, k_C} \mathbb{E}_{H_1}[\log(1 + \lambda V_t)],$$

which is the standard criterion for maximizing asymptotic wealth growth and is closely connected to powerful e-value constructions (see, e.g. Ramdas & Wang, 2025, Sections

**Algorithm 1** Sequential KCI Testing via Betting

1: **Input:** Initial training set $\mathcal{X}_0^{\text{tr}}$, initial validation set $\mathcal{X}_0^{\text{val}}$, stream of batches $\{\mathcal{Y}_t\}_{t=1}^{\infty}$ of size $b$, threshold $1/\alpha$, regularization $\varepsilon$.
2: **Initialize:** $W_0 \leftarrow 1$, $\eta_0 \leftarrow 0$, $\widehat{\gamma}_0 \leftarrow 0$, kernel $h^{(0)}$.
3: **for** $t = 1, 2, \ldots$ **do**
4:     {*Phase 1: Conditional Means Estimation*}
5:     Select kernel hyperparameters and fit $\mu_{A|C}^{(t)}$ and $\mu_{B|C}^{(t)}$ via kernel ridge regression on $\mathcal{X}_{t-1}^{\text{tr}}$.
6:     {*Phase 2: Strategy Optimization & Shift Estimation*}
7:     Set $\eta_t^{(0)} \leftarrow \eta_{t-1}$, $h^{(t,0)} \leftarrow h^{(t-1)}$, $\widehat{\gamma}_t^{(0)} \leftarrow \widehat{\gamma}_{t-1}$.
8:     **for** $s = 1, \ldots, S$ **do**
9:         Take a gradient step to update $\eta_t^{(s)}$ and $h^{(t,s)}$ (in the parameters of $k_C$ only) to maximize

$$\sum_{i=1}^{t} \log\left(1 + \sigma(\eta_t^{(s)}) \max\{\widetilde{V}_i^{(t,s)} - \widehat{\gamma}_t^{(s-1)}, -1\}\right)$$

        where historical payoff proxies $\{\widetilde{V}_i^{(t,s)}\}_{i=1}^{t}$ are based on the current kernel.
10:        Select the shift $\widehat{\gamma}_t^{(s)}$ for the new kernel via binary search using $\mathcal{X}_{\text{tr}}^{(t-1)}$ and $\mathcal{X}_{\text{val}}^{(t-1)}$.
11:     **end for**
12:     Set $\eta_t \leftarrow \eta_t^{(S)}$, $h^{(t)} \leftarrow h^{(t,S)}$, $\widehat{\gamma}_t \leftarrow \widehat{\gamma}_t^{(S)}$.
13:     {*Phase 3: Wealth Update*}
14:     Receive new test batch $\mathcal{Y}_t$.
15:     Compute $V_t^{\text{raw}} \leftarrow \frac{U_{n,b}^{(t)}}{S_n^{(t)} + \varepsilon}$ using $h^{(t)}$.
16:     Compute final payoff $V_t \leftarrow \max\{V_t^{\text{raw}} - \widehat{\gamma}_t, -1\}$.
17:     Update wealth $W_t \leftarrow W_{t-1}\left(1 + \sigma(\eta_t)V_t\right)$.
18:     **if** $W_t \geq 1/\alpha$ **then**
19:         **Reject $H_0$ and terminate.**
20:     **end if**
21:     $\mathcal{X}_{\text{tr}}^{(t)} \leftarrow \mathcal{X}_{\text{tr}}^{(t-1)} \cup \mathcal{X}_{\text{val}}^{(t-1)}, \quad \mathcal{X}_{\text{val}}^{(t)} \leftarrow \mathcal{Y}_t$
22: **end for**

3.3 and 7.8). Since the alternative distribution is unknown, we replace this population objective by an empirical proxy computed from the historical data available before round $t$.

At each round $t$, we compute proxy payoffs computed with the current kernel $h^{(t)}$. Since the observations are i.i.d., these proxy payoffs provide a pre-round estimate of how the current kernel and betting fraction would perform on fresh test batches.

To construct the proxy payoffs, we partition the $n$ historical training samples $\mathcal{X}_{t-1}^{\text{tr}}$ into blocks of size $b$, and write $\mathcal{I}_i$ for the indices in the $i$th block. Since data arrives in batches of size $b$, we have $t$ blocks in total. For each block, we treat the samples in $\mathcal{I}_i$ as a pseudo-test batch and compare them with the remaining historical samples.

Specifically, we define

$$\widetilde{V}_i^{(t)} = \frac{\sum_{l \in \mathcal{I}_i} \sum_{j \notin \mathcal{I}_i} h^{(t)}(x_j, x_l)}{b(n-b)\left(S_n^{(t)}(\mathcal{X}_{t-1}^{\text{tr}}) + \varepsilon\right)}, \quad i \in \{1, \ldots, t\}. \tag{9}$$

This leave-block-out construction avoids self-interaction terms and gives an empirical proxy for the current normalized payoff. We then select $k_C$ and $\lambda_t$ by maximizing the empirical log-wealth criterion over these proxy payoffs.

To enforce the constraint $\lambda_t \in (0,1)$, we parameterize $\lambda_t = \sigma(\eta_t)$ using the logistic sigmoid function, and jointly optimize it along with kernel parameters on our block estimate of the expected log-wealth growth:

$$\sum_{i=1}^{t} \log\left(1 + \sigma(\eta_t)\max\{\widetilde{V}_i^{(t)} - \gamma^{(t)}, -1\}\right). \tag{10}$$

The overall framework for our sequential testing procedure, which we call Sequential Kernel Conditional Independence (SKCI), is summarized in Algorithm 1.

## 4. Theoretical Analysis

We analyze the statistical validity of the proposed sequential test under the null hypothesis $H_0$. The main difficulty is that the wealth process $W_t$, constructed using the implemented Gaussian shift $\widehat{\gamma}_t$, is not an exact null supermartingale. We quantify the deviation from the ideal supermartingale property through the conditional drift

$$\delta_t := \mathbb{E}_{H_0}[\max\{V_t^{\text{raw}} - \widehat{\gamma}_t, -1\} \mid \mathcal{F}_{t-1}]. \tag{11}$$

If $\delta_t \leq 0$, the one-step wealth update is conservative under $H_0$, so we derive an explicit deterministic upper bound on $\delta_t$ and use it to construct a compensated supermartingale.

The full proof is given in Appendix A. We summarize the argument in three steps: first, we establish the sensitivity of the Gaussian null-calibrating shift; second, we combine this sensitivity bound with a finite-sample Gaussian approximation to obtain the one-step drift bound; third, we convert the one-step drift bound into the Type I error bound.

### 4.1. Sensitivity of the Null-Calibrating Shift

We define the Gaussian-calibrated shift using the Gaussian approximation to the conditional null law of $V_t^{\text{raw}}$. For parameters $(\mu, \sigma)$, let

$$f(\gamma; \mu, \sigma) := \mathbb{E}_{Z \sim \mathcal{N}(\mu, \sigma^2)}\big[\max\{Z - \gamma, -1\}\big].$$

The Gaussian calibrating shift $\gamma(\mu, \sigma)$ is defined by

$$f(\gamma(\mu, \sigma); \mu, \sigma) = 0.$$

In particular, $\gamma(\mu_t, \sigma_t)$ denotes the Gaussian shift corresponding to the limiting approximation $N(\mu_t, \sigma_t^2)$ of

$V_t^{\text{raw}} \mid \mathcal{F}_{t-1}$, while the implemented shift $\widehat{\gamma}_t$ is selected using the centered Gaussian law $N(0, \widehat{\sigma}_t^2)$.

The following lemma controls the effect of using the wrong Gaussian mean and variance.

**Lemma 4.1** (Implicit Sensitivity). *The map* $(\mu, \sigma) \mapsto \gamma(\mu, \sigma)$ *is continuously differentiable. Moreover, for any* $(\mu, \sigma)$ *and* $(\widehat{\mu}, \widehat{\sigma})$,

$$\gamma(\mu, \sigma) - \gamma(\widehat{\mu}, \widehat{\sigma}) \leq \mu - \widehat{\mu} + \Lambda(\sigma)(\sigma - \widehat{\sigma}),$$

*where* $\Lambda(\sigma) := \frac{\phi(\xi_\sigma)}{\Phi(-\xi_\sigma)}$, *and* $\xi_\sigma$ *is the unique solution of* $\sigma\{\phi(\xi) - \xi\Phi(-\xi)\} = 1$. *Here* $\phi$ *and* $\Phi$ *denote the standard normal density and distribution function.*

This lemma allows us to convert mean and variance mismatch in the Gaussian calibration law into an upper bound on the excess expected payoff.

### 4.2. Drift Decomposition

The drift has two sources: a finite-block Gaussian approximation gap for $V_t^{\text{raw}} \mid \mathcal{F}_{t-1}$, and a Gaussian calibration mismatch from using $\widehat{\gamma}_t$, calibrated under $N(0, \widehat{\sigma}_t^2)$ instead $N(\mu_t, \sigma_t^2)$. Let $\delta_{A|C}^{(t)}$ and $\delta_{B|C}^{(t)}$ denote the conditional mean embedding regression errors at step $t$.

**Theorem 4.2** (Drift Upper Bound). *Under* $H_0$, *assume the kernel is bounded such that* $\sup_x h^{(t)}(x, x) \leq \kappa$, *and* $\sup_c \|\phi_C(c)\| \leq 1$. *Assume that the conditional absolute third central moment of* $\frac{1}{n}\sum_{i=1}^n h^{(t)}(x_i, Y)$ *is uniformly bounded and that its conditional variance is uniformly non-degenerate, so that its Berry–Esseen moment ratio is at most* $\rho$. *Then for each* $t \leq T$, *the drift* $\delta_t$ *of* (11) *satisfies*

$$\delta_t \leq U_t := \frac{C_1 \rho}{b\,\varepsilon} + \frac{\sqrt{\kappa}}{\varepsilon}\|\delta_{A|C}^{(t)}\|\|\delta_{B|C}^{(t)}\| + \frac{2C_2\kappa^2}{b\,\varepsilon^2}.$$

The first term comes from the finite-block Gaussian approximation gap, which we control using a Wasserstein Berry–Esseen bound for $V_t^{\text{raw}}$; here $C_1 > 0$ is a universal constant. The second and third terms arise from the Gaussian calibration mismatch controlled by Lemma 4.1. The second term captures the effect of the nonzero conditional mean $\mu_t$, which is induced by conditional mean embedding estimation error and vanishes when the residualizers recover the population conditional mean embeddings. The third term controls the mismatch between the Gaussian variance parameter $\sigma_t^2$ and the implemented variance estimate $\widehat{\sigma}_t^2$; here $C_2 > 0$ is another universal constant.

The ablations in Figure 12 are consistent with this drift upper bound: larger batch sizes $b$ and regularization parameters $\varepsilon$ tend to reduce empirical null rejection rate, as suggested by the corresponding terms in $U_t$.

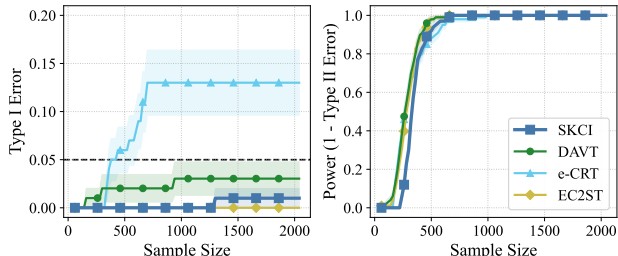

*Figure 1.* Linearly dependent Gaussian data in online mode.

### 4.3. Finite-Sample Type I Error Inflation

The following result translates a one-step drift bound, such as Theorem 4.2, into a Type I error bound. Since $\mathbb{E}_{H_0}[V_t \mid \mathcal{F}_{t-1}] = \delta_t \leq U_t$, the corrected process $\widetilde{W}_t := \frac{W_t}{\prod_{i=1}^t (1+\lambda_i U_i)}$ is a nonnegative supermartingale under $H_0$, and Ville's inequality yields the following result.

**Proposition 4.3** (Finite-Sample Type I Error). *Let* $\alpha \in (0, 1)$. *Under* $H_0$, *suppose that for all* $t$, $\delta_t \leq U_t$. *Then a test rejecting when* $W_t \geq \frac{1}{\alpha}\prod_{i=1}^t (1 + \lambda_i U_i)$ *has anytime control of its Type I error at level* $\alpha$. *This also implies that the original test satisfies*

$$\Pr_{H_0}\left(\exists t \leq T : W_t \geq \frac{1}{\alpha}\right) \leq \alpha \exp\left(\sum_{t=1}^T \lambda_t U_t\right).$$

## 5. Experiments

We evaluate the SKCI betting framework across synthetic and real-world benchmarks designed to evaluate both anytime Type I error control and power for difficult problems.

Across all experiments, we compare SKCI against e-CRT (Shaer et al., 2023), DAVT (Pandeva et al., 2024b), and a version of EC2ST (Pandeva et al., 2024a) that distinguish true $(A, B, C)$ triples from $(\widetilde{A}, B, C)$ knockoffs. We use a common batch size $b = 20$, and average the results over 100 independent runs unless otherwise specified. Shaded regions in all plots indicate the standard error. We use the same regression architecture for baseline methods for fair comparison; details in Appendix B.4.

To isolate the effect of estimation quality, we report results in three regimes. (i) In Oracle mode, the conditional distribution $P_{A|C}$ is known exactly; this is possible for synthetic data only. (ii) In Pretrained mode, $P_{A|C}$ is estimated from a large offline dataset (we use 3,000 samples). (iii) In Online mode, no prior side data is available, and conditional mean embeddings are updated sequentially as we see data.

Due to space constraints, Oracle results, Pretrained results and additional configurations and ablation studies are deferred to Appendix C.

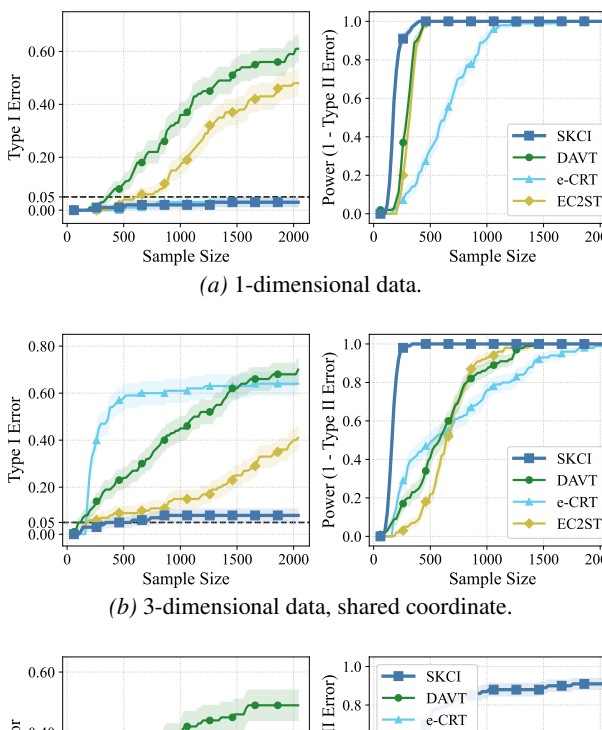

*(a)* 1-dimensional data.

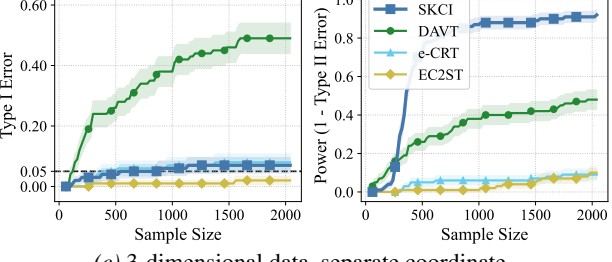

*(b)* 3-dimensional data, shared coordinate.

*(c)* 3-dimensional data, separate coordinate.

*Figure 2.* CI hardness data (He et al., 2025) in Online mode.

**Linearly Dependent Gaussian Data** We begin with a Gaussian benchmark commonly used in recent work on model-free conditional independence testing (Shaer et al., 2023; Pandeva et al., 2024b). We sample $C \sim \mathcal{N}(0, \mathbf{I}_{19})$, $u \sim \mathcal{N}(0, \mathbf{I}_{19})$, and $A \mid (C, u) \sim \mathcal{N}(u^\top C, 1)$. Under the null hypothesis, we define $B \mid (A, C) \sim \mathcal{N}((w^\top C)^2, 1)$, so that $A \perp\!\!\!\perp B \mid C$ despite having strong nonlinear dependence on $C$. Under the alternative, we introduce a linear dependence on $A$: $B \mid (A, C) \sim \mathcal{N}((w^\top C)^2 + 3A, 1)$.

Figure 1, as well as Figure 6 in the appendix, show Type I and power curves. In Oracle mode, all methods control Type I error and have reasonable power. In Pretrained and Online modes, some baseline methods begin to suffer from worse Type I error rates, while SKCI remains stable and rapidly achieves high power.

**CI Hardness Benchmarks (1D & 3D)** The conditional independence hardness benchmarks of He et al. (2025) have $C$-varying dependence which is particularly difficult to detect. Here $C \sim \mathcal{N}(0, I)$, $A = \cos(e_A^\top C) + 0.1 r_A$, and $B = \exp(e_B^\top C) + 0.1 r_B$, where $(r_A, r_B)$ are jointly Gaus-

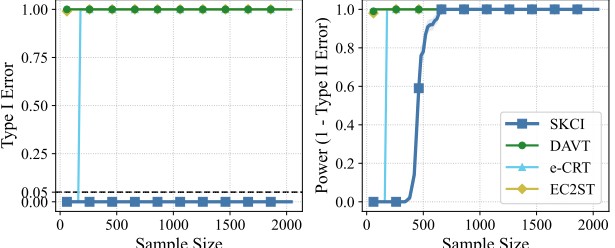

*Figure 3.* Synthetic neural data in Online mode.

sian with unit variance and conditional covariance $\gamma(e_C^\top C)$. Under $H_0$, $\gamma(c) = 0$; under $H_1$, $\gamma(c) = \sin(3c)$.

There are three configurations, of increasing difficulty: (i) 1D, where $C \in \mathbb{R}$ and $e_A = e_B = e_C = 1$ (Figures 2a, 7a and 7b); (ii) 3D with shared coordinates, $e_A = e_B = e_C$ (Figures 2b, 8a and 8b); (iii) 3D with separate coordinates, where $e_A$, $e_B$, $e_C$ are orthonormal (Figures 2c, 9a and 9b).

In the 1D and shared-coordinate settings, SKCI consistently matches or exceeds the power of competing methods while maintaining tight Type I control. In Oracle mode, all methods maintain controlled Type I error, but SKCI achieves the highest power; other methods struggle with Type I error and/or power in Pretrained and particularly Online mode. The separate-coordinate 3D setting is substantially harder: in all three modes, other methods fail to detect dependence and/or suffer severe Type I error inflation. In contrast, SKCI's online kernel optimization successfully adapts to the relevant subspace, yielding reliable power and reasonable Type I control even when the dependence signal is decoupled from marginal structure.

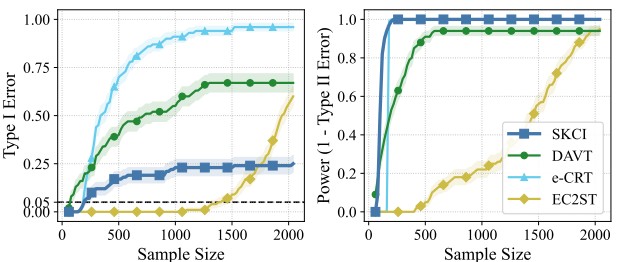

*Figure 4.* dSprites data, Online mode.

**Synthetic Neural Data (RatInABox)** Following Pogodin et al. (2024), we evaluate SKCI in a high-dimensional, biologically motivated setting using the RatInABox simulator (George et al., 2024). The goal is to test whether head-direction cells ($A \in \mathbb{R}^{100}$) – neurons that fire as a function of the animal's heading – are conditionally independent of conjunctive cells ($B \in \mathbb{R}^{100}$) – which respond jointly to heading and spatial location – given the animal's physical state $C \in \mathbb{R}^4$ (position and orientation). That is, we ask

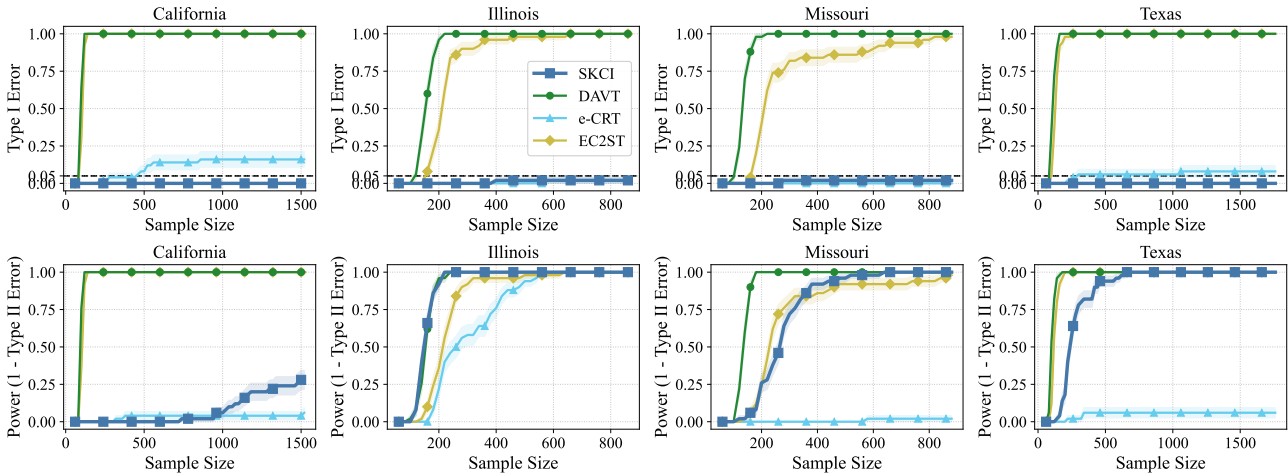

*Figure 5.* Car insurance discrimination data, Online mode. Top rows give Type I error, bottom rows give power for data across four states.

whether conjunctive activity is computed "downstream" of head-direction signals (our $H_1$) or is computed separately (our $H_0$), despite common dependence on $C$.

Figures 3 and 10 reports results; exact distributions for Oracle mode are intractable. In this challenging high-dimensional regime, SKCI achieves strong power while tightly controlling Type I error. EC2ST suffers extreme Type I error, as does DAVT in Online mode; DAVT in Pre-trained achieves almost no power. e-CRT has good Type I control but is much slower to detect dependence than SKCI.

**Image Data (dSprites)** Inspired by Zhang et al. (2025), we construct an image-based conditional independence task from dSprites (Matthey et al., 2017). Let $A$ denote the shape of the object, $B$ denote the full image, and $C$ denote a cropped view of $B$. When the cropped image $C$ contains only part of the object, the full image $B$ carries additional information about the shape beyond $C$, so $A \not\!\perp\!\!\!\perp B \mid C$. If $C$ contains the full object, $B$ provides no additional shape information, and $A \perp\!\!\!\perp B \mid C$.

Figure 4 shows that SKCI maintains Type I error substantially better than the competing baselines, whose rejection rates quickly approach one even when the crop already contains the full object. In the dependent setting, all methods achieve high power with moderate sample sizes, indicating that the main distinction on this task is null calibration rather than sensitivity to the alternative.

**Car Insurance Discrimination** Finally, we apply SKCI to the car insurance dataset of Angwin et al. (2017), following the auditing protocol of Polo et al. (2023) and Pogodin et al. (2024). We test whether insurance premiums ($A$) are conditionally independent of minority neighborhood status ($B$), given driver risk factors ($C$). We only consider On-

line mode, since data sizes are not large enough to support significant splits for Pretrained models.

To assess Type I error, we construct a synthetic null by clustering $C$ and shuffling premiums within clusters, stratified by company and state. The results are computed from 50 runs per (state, company) pair, and the final decision is obtained via majority vote across companies within each state. Figure 5 reports rejection rates and power across four U.S. states in Online mode. SKCI maintains conservative Type I error across all states while achieving competitive or superior power. DAVT and EC2ST have Type I error rates near 1, while e-CRT again is more controlled but underpowered.

## 6. Discussion

While exact uniform Type I control is impossible without assumptions, our theory shows that SKCI's Type I inflation can be controlled over finite time horizons when the conditional mean embedding estimates are sufficiently accurate. Although existing theoretical rates for conditional mean embedding estimation can be slow (Li et al., 2022; 2024), we find that, across several challenging synthetic and real-world benchmarks, SKCI tends to exhibit substantially better Type I behavior than prior methods while retaining competitive power for detecting conditional dependence.

## Acknowledgements

The authors would like to thank Nathaniel Xu and the anonymous reviewers for several useful suggestions.

This work was enabled in part by support from the Natural Sciences and Engineering Research Council of Canada, the Canada CIFAR AI Chairs Program, Calcul Québec, the BC DRI Group, and the Digital Research Alliance of Canada.

## Impact Statement

Conditional independence testing is a tool with broad applications across machine learning and statistics, including some that are good for society and others that are more dubious. We do not feel our work, which helps increase the reliability of an existing class of methods, raises any particular consequences that must be discussed here.

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

## A. Proof of Type I Error Bound

We prove the finite-sample Type I error bound for the sequential test under $H_0$. Recall that the wealth process is updated as

$$W_t = W_{t-1}(1 + \lambda_t \widehat{V}_t), \qquad \widehat{V}_t = \max\{V_t^{\text{raw}} - \widehat{\gamma}_t, -1\},$$

where $\widehat{\gamma}_t$ is the implemented Gaussian calibration shift. If $\widehat{\gamma}_t$ were the exact null-calibrating shift for the conditional law of $V_t^{\text{raw}} \mid \mathcal{F}_{t-1}$, then the conditional expected payoff would be zero and the resulting wealth process would be an exact martingale. In finite samples, however, $\widehat{\gamma}_t$ is only an approximate shift, so the wealth process may have a nonzero conditional drift.

We define this drift by

$$\delta_t := \mathbb{E}_{H_0}[\max\{V_t^{\text{raw}} - \widehat{\gamma}_t, -1\} \mid \mathcal{F}_{t-1}].$$

Throughout this section, all expectations and distributions are conditional on $\mathcal{F}_{t-1}$ unless stated otherwise.

The proof proceeds in four parts. First, we decompose $\delta_t$ into a finite-block Gaussian approximation term and a Gaussian calibration mismatch term. Second, we establish the sensitivity of the Gaussian null-calibrating shift with respect to its mean and variance parameters. Third, we bound the mean and variance mismatch induced by conditional mean embedding estimation and by the implemented variance scale. Finally, we combine these bounds to obtain a deterministic one-step drift bound and use it to construct a compensated supermartingale, from which the finite-sample Type I error bound follows by Ville's inequality.

### A.1. Decomposition of the Supermartingale Drift

In this section, we decompose the conditional drift of the betting payoff under the null. The payoff depends on the null-calibrating shift $\widehat{\gamma}_t$, which is selected using the Gaussian approximation $N(\mu_t, \sigma_t^2)$. The implemented statistic is based on a finite block of null-calibrating samples and on the data-adaptive kernel $h^{(t)}$, whose construction involves estimated conditional mean embeddings. We therefore separate the drift into a Gaussian calibration term and a residual approximation term. This decomposition allows us to identify the conditions under which the betting process remains close to a null supermartingale.

Let $\overline{\mathbb{E}}_{H_0}[\cdot]$ and $\overline{\text{Var}}_{H_0}[\cdot]$ denote conditional expectation and variance under the null distribution given the filtration $\mathcal{F}_{t-1}$. Define the conditional drift of the truncated payoff by

$$\delta_t := \mathbb{E}_{H_0}\big[\max\{V_t^{\text{raw}} - \widehat{\gamma}_t, -1\} \mid \mathcal{F}_{t-1}\big] = \overline{\mathbb{E}}_{H_0}\big[\max\{V_t^{\text{raw}} - \widehat{\gamma}_t, -1\}\big].$$

We decompose this drift by adding and subtracting the corresponding expectation under the Gaussian approximation. Let $X_t$ denote a conditionally Gaussian random variable such that

$$X_t \mid \mathcal{F}_{t-1} \sim N(\mu_t, \sigma_t^2).$$

Then

$$\delta_t = \underbrace{\overline{\mathbb{E}}_{H_0}\big[\max\{V_t^{\text{raw}} - \widehat{\gamma}_t, -1\}\big] - \overline{\mathbb{E}}_{X_t \sim N(\mu_t, \sigma_t^2)}\big[\max\{X_t - \widehat{\gamma}_t, -1\}\big]}_{\text{(I) Gaussian approximation error}} + \underbrace{\overline{\mathbb{E}}_{X_t \sim N(\mu_t, \sigma_t^2)}\big[\max\{X_t - \widehat{\gamma}_t, -1\}\big]}_{\text{(II) Gaussian calibration error}}. \quad (12)$$

The first term measures the discrepancy between the true conditional null law of $V_t^{\text{raw}}$ and its limiting Gaussian law $N(\mu_t, \sigma_t^2)$. This term captures finite-sample effects. The second term is the drift obtained when the implemented shift $\widehat{\gamma}_t$ is evaluated under this Gaussian limiting law. Since $\widehat{\gamma}_t$ is selected using the centered calibration law $\mathcal{N}(0, \widehat{\sigma}_t^2)$, this term reflects the calibration mismatch between $N(0, \widehat{\sigma}_t^2)$ and $N(\mu_t, \sigma_t^2)$. The ideal finite-sample shift $\gamma_t^*$ would instead be calibrated against the true conditional null law of $V_t^{\text{raw}} \mid \mathcal{F}_{t-1}$. The decomposition above isolates the error due to this approximation from the error due to the choice of the Gaussian-calibrated shift $\widehat{\gamma}_t$.

We now describe the Gaussian approximation used in (12). Conditional on $\mathcal{F}_{t-1}$, the historical training samples $\{x_i\}_{i=1}^n \subseteq \mathcal{X}_{t-1}^{\text{tr}}$, the statistic $S_n(\mathcal{X}_{t-1}^{\text{tr}})$, and the kernel $h^{(t)}$ are fixed. For a fresh null sample $y_j$, define

$$g^{(t)}(y_j) := \frac{1}{n\big(S_n(\mathcal{X}_{t-1}^{\text{tr}}) + \varepsilon\big)} \sum_{i=1}^n h^{(t)}(x_i, y_j).$$

Then the raw betting statistic can be written as the empirical average

$$V_t^{\text{raw}} = \frac{1}{b} \sum_{j=1}^{b} g^{(t)}(y_j).$$

Under the null and conditional on $\mathcal{F}_{t-1}$, the variables $\{g^{(t)}(y_j)\}_{j=1}^{b}$ are i.i.d. We define their conditional mean and variance by

$$m_t := \bar{\mathbb{E}}_{H_0}\big[g^{(t)}(Y)\big], \qquad s_t^2 := \overline{\text{Var}}_{H_0}\big(g^{(t)}(Y)\big),$$

where $Y$ denotes a fresh null draw. Consequently,

$$\bar{\mathbb{E}}_{H_0}[V_t^{\text{raw}}] = m_t, \qquad \overline{\text{Var}}_{H_0}(V_t^{\text{raw}}) = \frac{s_t^2}{b}.$$

Thus the natural Gaussian approximation is

$$V_t^{\text{raw}} \mid \mathcal{F}_{t-1} \approx \mathcal{N}\left(m_t, \frac{s_t^2}{b}\right).$$

Equivalently, in the notation of (12),

$$\mu_t = m_t, \qquad \sigma_t^2 = \frac{s_t^2}{b}.$$

By the conditional central limit theorem, provided $0 < s_t^2 < \infty$,

$$\frac{\sqrt{b}(V_t^{\text{raw}} - m_t)}{s_t} \xrightarrow{d} \mathcal{N}(0,1) \qquad \text{conditionally on } \mathcal{F}_{t-1}.$$

Therefore, for large $b$, the conditional null law of $V_t^{\text{raw}} \mid \mathcal{F}_{t-1}$ is approximated by $\mathcal{N}(\mu_t, \frac{s_t^2}{b})$, the same conditional Gaussian law used to define $X_t$.

**Bounding Term I (Asymptotic Gap)** We bound the Asymptotic Gap using the Wasserstein metric. Define the test function $\ell(v) := \max\{v - \widehat{\gamma}_t, -1\}$.

**Lemma A.1** (Lipschitz Continuity of the Payoff). *The function $\ell(v)$ is Lipschitz continuous with constant $L = 1$.*

*Proof.* For any $x, y \in \mathbb{R}$,

$$|\ell(x) - \ell(y)| = |\max\{x - \widehat{\gamma}_t, -1\} - \max\{y - \widehat{\gamma}_t, -1\}| \le |(x - \widehat{\gamma}_t) - (y - \widehat{\gamma}_t)| = |x - y|. \qquad \square$$

By Kantorovich-Rubinstein duality, the Wasserstein-1 distance between distributions $P$ and $Q$ is given by

$$W_1(P, Q) = \sup_{f \in \text{Lip}(1)} \left|\mathbb{E}_{X \sim P}[f(X)] - \mathbb{E}_{Y \sim Q}[f(Y)]\right|.$$

Since $\ell \in \text{Lip}(1)$, we immediately have that

$$|\text{Term (I)}| = \left|\bar{\mathbb{E}}_{H_0}[\ell(V_t^{\text{raw}})] - \bar{\mathbb{E}}_{\mathcal{N}(\mu_t, \sigma_t^2)}[\ell(X_t)]\right| \le W_1\left(P_{H_0}(V_t^{\text{raw}} \in \cdot \mid \mathcal{F}_{t-1}), \mathcal{N}(\mu_t, \sigma_t^2)\right). \tag{13}$$

Conditioning on $\mathcal{F}_{t-1}$, let

$$Z_t := \frac{1}{n} \sum_{i=1}^{n} h^{(t)}(x_i, Y).$$

By Berry-Esseen-type Wasserstein bounds, such as Corollary 4.2 of Chen et al. (2010),

$$W_1\left(P_{H_0}(V_t^{\text{raw}} \in \cdot \mid \mathcal{F}_{t-1}), \mathcal{N}(\mu_t, \sigma_t^2)\right) \le \frac{C_1 \rho_t}{b\big(S_n(\mathcal{X}_{t-1}^{\text{tr}}) + \varepsilon\big)} \le \frac{C_1 \rho_t}{b\varepsilon},$$

where the Berry–Esseen moment ratio is defined as

$$\rho_t := \frac{\bar{\mathbb{E}}_{H_0}\big[|Z_t - \bar{\mathbb{E}}_{H_0}[Z_t]|^3\big]}{\overline{\mathrm{Var}}_{H_0}(Z_t)},$$

and $C_1 > 0$ is a universal constant. Since we assume that the conditional third central moment of $Z$ is uniformly bounded and that its conditional variance is uniformly nondegenerate, the Berry–Esseen moment ratio $\rho_t$ is uniformly bounded. We therefore assume that there exists $\rho < \infty$ such that $\rho_t \leq \rho$ for all $t \leq T$. Thus,

$$|\text{Term (I)}| \leq \frac{C_1 \rho}{b\,\varepsilon}. \tag{14}$$

**Term II: Calibration mismatch.** The second term is the drift obtained when the implemented shift $\widehat{\gamma}_t$ is evaluated under the Gaussian approximation to the conditional null law of $V_t^{\mathrm{raw}}$. This term is zero or negative if $\widehat{\gamma}_t$ is at least as large as the ideal calibrating shift for this Gaussian law. Therefore, bounding this term reduces to controlling the difference between the implemented shift $\widehat{\gamma}_t$, selected using $N(0, \widehat{\sigma}_t^2)$, and the Gaussian-calibrating shift associated with $N(\mu_t, \sigma_t^2)$.

We defer the quantitative bound to Appendix A.2, where we introduce the Gaussian payoff function

$$f(\gamma; \mu, \sigma) := \mathbb{E}_{X \sim N(\mu, \sigma^2)}\big[\max\{X - \gamma, -1\}\big].$$

Denote $\gamma(\mu_t, \sigma_t)$ the Gaussian calibrating shift for $N(\mu_t, \sigma_t^2)$. With this notation,

$$\begin{aligned}
\text{Term (II)} &= \bar{\mathbb{E}}_{X_t \sim N(\mu_t, \sigma_t^2)}\big[\max\{X_t - \widehat{\gamma}_t, -1\}\big] \\
&= f(\widehat{\gamma}_t; \mu_t, \sigma_t) - \underbrace{f(\gamma(\mu_t, \sigma_t); \mu_t, \sigma_t)}_{=0}.
\end{aligned}$$

This term is controlled by the discrepancy between $\widehat{\gamma}_t$ and $\gamma(\mu_t, \sigma_t)$. In Appendix A.2, we bound the sensitivity of the calibrating map $(\mu, \sigma) \mapsto \gamma(\mu, \sigma)$. Combining this sensitivity bound with the mean and variance mismatch bounds from Appendix A.3 gives the desired bound on Term (II).

### A.2. Sensitivity of the Null-Calibrating Shift

We will use $\mathcal{N}(\mu, \sigma^2)$ to denote a Gaussian distribution with mean $\mu$ and variance $\sigma^2$, $\Phi(\cdot; \mu, \sigma^2)$ for its CDF, and $\phi(\cdot; \mu, \sigma^2)$ for its PDF. $\Phi(\cdot)$ and $\phi(\cdot)$ refer to the case where $\mu = 0$, $\sigma^2 = 1$.

Define the functional $f : \mathbb{R} \times \mathbb{R} \times \mathbb{R}_{>0} \to \mathbb{R}$ representing the expected truncated payoff under a Gaussian law by

$$f(\gamma; \mu, \sigma) := \mathbb{E}_{X \sim \mathcal{N}(\mu, \sigma^2)}\big[\max\{X - \gamma, -1\}\big]. \tag{15}$$

The function $f$ is strictly decreasing in $\gamma$. Therefore, the ideal null-calibrating shift $\gamma(\mu, \sigma)$ for a random variable $X \sim \mathcal{N}(\mu, \sigma^2)$ is defined as the unique solution to $f(\gamma; \mu, \sigma) = 0$.

**Lemma A.2** (Implicit Sensitivity of the Shift). *The mapping $(\mu, \sigma) \mapsto \gamma(\mu, \sigma)$ is continuously differentiable on $\mathbb{R} \times \mathbb{R}_{>0}$ and satisfies the following properties:*

- *Mean sensitivity:* $\frac{\partial \gamma}{\partial \mu} = 1$.

- *Variance sensitivity:* *The variance derivative $\frac{\partial \gamma}{\partial \sigma}$ is both positive and monotonically increasing in $\sigma$.*

- *Lipschitz bound:* *For any $(\mu_1, \sigma_1)$ and $(\mu_2, \sigma_2)$,*

$$\gamma(\mu_1, \sigma_1) - \gamma(\mu_2, \sigma_2) \leq (\mu_1 - \mu_2) + \Lambda(\sigma_1)(\sigma_1 - \sigma_2),$$

  *where $\Lambda(\sigma) := \frac{\phi(\xi_\sigma)}{\Phi(-\xi_\sigma)}$ and $\xi_\sigma$ is the unique solution to $\sigma\,[\phi(\xi) - \xi\Phi(-\xi)] = 1$.*

*Proof.* Since the expectation involves truncation at $-1$, the function $f$ admits the representation

$$f(\gamma; \mu, \sigma) = \int_{-\infty}^{\gamma - 1} (-1)\, \phi(v; \mu, \sigma)\, dv + \int_{\gamma - 1}^{\infty} (v - \gamma)\, \phi(v; \mu, \sigma)\, dv$$

$$= \sigma\, \phi\left(\frac{\gamma - \mu - 1}{\sigma}\right) - (\gamma - \mu - 1)\, \Phi\left(-\frac{\gamma - \mu - 1}{\sigma}\right) - 1.$$

The function $f$ is continuously differentiable and strictly decreasing in $\gamma$. Its partial derivatives are

$$\frac{\partial f}{\partial \gamma} = -\Phi\left(-\frac{\gamma - \mu - 1}{\sigma}\right), \quad \frac{\partial f}{\partial \mu} = \Phi\left(-\frac{\gamma - \mu - 1}{\sigma}\right), \quad \frac{\partial f}{\partial \sigma} = \phi\left(\frac{\gamma - \mu - 1}{\sigma}\right).$$

Let $\xi := (\gamma(\mu, \sigma) - \mu - 1)/\sigma$. Since $\partial f/\partial \gamma \in (-1, 0)$ for all $\mu$ and all $\sigma > 0$, the implicit function theorem guarantees that $\gamma(\mu, \sigma)$ is continuously differentiable, with

$$\frac{\partial \gamma}{\partial \mu} = -\frac{\partial f/\partial \mu}{\partial f/\partial \gamma} = 1, \qquad \frac{\partial \gamma}{\partial \sigma} = -\frac{\partial f/\partial \sigma}{\partial f/\partial \gamma} = \frac{\phi(\xi)}{\Phi(-\xi)}. \tag{16}$$

From the defining equation $f(\gamma; \mu, \sigma) = 0$, we obtain

$$1 = \sigma\left[\phi(\xi) - \xi\Phi(-\xi)\right] \quad \Longleftrightarrow \quad \frac{1}{\sigma} = r(\xi),$$

where $r(\xi) := \phi(\xi) - \xi\Phi(-\xi)$. Since $r'(\xi) = -\Phi(-\xi) < 0$, the function $r$ is strictly decreasing in $\xi$. Hence the solution $\xi_\sigma$ to $1/\sigma = r(\xi)$ is unique and is strictly increasing in $\sigma$.

Moreover, the ratio $\phi(\xi)/\Phi(-\xi)$ is strictly increasing in $\xi$. Since $\partial \gamma/\partial \sigma = \phi(\xi)/\Phi(-\xi)$, it follows that $\partial \gamma/\partial \sigma$ is strictly increasing in $\xi$.

Let $\xi_\sigma$ denote the unique solution to $1/\sigma = r(\xi)$, so that $\xi$ is implicitly determined by $\sigma$. Define

$$\Lambda(\sigma) := \frac{\partial \gamma}{\partial \sigma} = \frac{\phi(\xi_\sigma)}{\Phi(-\xi_\sigma)}.$$

Then, since $\xi_\sigma$ is strictly increasing in $\sigma$ and $\phi(\xi)/\Phi(-\xi)$ is strictly increasing in $\xi$, it follows that $\Lambda(\sigma)$ is strictly increasing in $\sigma$.

By the mean value theorem, there exists $(\tilde{\mu}, \tilde{\sigma})$ on the line segment connecting $(\mu_1, \sigma_1)$ and $(\mu_2, \sigma_2)$ such that

$$\gamma(\mu_1, \sigma_1) - \gamma(\mu_2, \sigma_2) = (\mu_1 - \mu_2) + \Lambda(\tilde{\sigma})(\sigma_1 - \sigma_2).$$

If $\sigma_1 \geq \sigma_2$, then $\tilde{\sigma} \leq \sigma_1$ and hence $\Lambda(\tilde{\sigma}) \leq \Lambda(\sigma_1)$. If $\sigma_1 < \sigma_2$, then $\sigma_1 - \sigma_2 < 0$ and $\Lambda(\tilde{\sigma}) > \Lambda(\sigma_1)$, which again yields $\Lambda(\tilde{\sigma})(\sigma_1 - \sigma_2) \leq \Lambda(\sigma_1)(\sigma_1 - \sigma_2)$. In both cases, the claimed bound follows. $\square$

We have shown that $\frac{\partial f}{\partial \gamma} \in (-1, 0)$ in the proof of Lemma A.2. Let $\gamma(\mu_t, \sigma_t)$ be the unique solution to $f(\gamma; \mu_t, \sigma_t) = 0$. When $\gamma(\mu_t, \sigma_t) \geq \widehat{\gamma}_t$, then

$$f(\widehat{\gamma}_t; \mu_t, \sigma_t) - f(\gamma(\mu_t, \sigma_t); \mu_t, \sigma_t) \leq \gamma(\mu_t, \sigma_t) - \widehat{\gamma}_t.$$

When $\gamma(\mu_t, \sigma_t) < \widehat{\gamma}_t$, then

$$f(\widehat{\gamma}_t; \mu_t, \sigma_t) - f(\gamma(\mu_t, \sigma_t); \mu_t, \sigma_t) \leq 0.$$

Thus,

$$\text{Term II} \leq \max\{\gamma(\mu_t, \sigma_t) - \widehat{\gamma}_t, 0\}.$$

Recall that $\widehat{\gamma}_t$ is selected using the Gaussian law $N(0, \widehat{\sigma}_t^2)$. Combining this with the Lipschitz bound from Lemma A.2 gives

$$\text{Term II} \leq \max\{\mu_t + \Lambda(\sigma_t)(\sigma_t - \widehat{\sigma}_t), 0\}.$$

This bound controls the calibration error by the mean and variance mismatch between the Gaussian approximation to the conditional null law of $V_t^{\text{raw}}$ and the Gaussian law used to select the implemented shift $\widehat{\gamma}_t$.

## A.3. Bounding the Mean and Variance Mismatch

In this section, we bound the mismatch between the Gaussian approximation $N(\mu_t, \sigma_t^2)$ to the conditional null law of $V_t^{\mathrm{raw}}$ and the centered Gaussian law $N(0, \widehat{\sigma}_t^2)$ used to select the implemented shift $\widehat{\gamma}_t$. Here $\mu_t$ and $\sigma_t^2$ denote the scalar mean and variance parameters of the Gaussian approximation to $V_t^{\mathrm{raw}} \mid \mathcal{F}_{t-1}$. They are distinct from the conditional mean embeddings $\mu_{A|C}$ and $\mu_{B|C}$, which are Hilbert-space valued functions used to construct the residualized kernel $h^{(t)}$.

We control this mismatch through two quantities:

$$|\mu_t| \qquad \text{and} \qquad |\sigma_t^2 - \widehat{\sigma}_t^2|.$$

The first term captures the nonzero mean induced by conditional mean embedding estimation error, while the second captures both conditional mean embedding error and finite-batch variation in the variance estimate.

Define the regression errors

$$\delta_{A|C}^{(t)}(c) := \mu_{A|C}^{(t)}(c) - \mu_{A|C}(c), \qquad \delta_{B|C}^{(t)}(c) := \mu_{B|C}^{(t)}(c) - \mu_{B|C}(c).$$

**Lemma A.3** (Bound on the Mean Mismatch)**.** *Assume the kernel is bounded such that* $\sup_x h^{(t)}(x, x) \leq \kappa$, *and* $\sup_c \|\phi_C(c)\| \leq 1$. *Then*

$$|\mu_t| \leq \frac{\sqrt{\kappa}}{\varepsilon} \|\delta_{A|C}^{(t)}\| \|\delta_{B|C}^{(t)}\|.$$

*Proof.* For simplicity, we write $S_n(\mathcal{X}_{t-1}^{\mathrm{tr}})$ as $S_n^{(t-1)}$. Recall that

$$\mu_t = \bar{\mathbb{E}}_{H_0}[V_t^{\mathrm{raw}}] = \frac{1}{S_n^{(t-1)} + \varepsilon} \bar{\mathbb{E}}_{H_0} \left[ \frac{1}{n} \sum_{i=1}^n h^{(t)}(x_i, Y) \right]. \tag{17}$$

Here $x_i = (a_i, b_i, c_i)$ are fixed training samples conditional on $\mathcal{F}_{t-1}$, while $Y = (A, B, C)$ denotes a fresh null draw.

Using the feature representation

$$h^{(t)}(x_i, Y) = \left\langle \psi^{(t)}(x_i), \psi^{(t)}(Y) \right\rangle_{\mathcal{H}},$$

where

$$\psi^{(t)}(x) := \left( \phi_A(a) - \mu_{A|C}^{(t)}(c) \right) \otimes \left( \phi_B(b) - \mu_{B|C}^{(t)}(c) \right) \otimes \phi_C(c),$$

is in the space of Hilbert-Schmidt (HS) operators $\mathcal{H} = \mathrm{HS}(\mathcal{H}_\mathcal{C}, \mathrm{HS}(\mathcal{H}_\mathcal{B}, \mathcal{H}_\mathcal{A}))$.

Expanding

$$\phi_A(A) - \mu_{A|C}^{(t)}(C) = \left( \phi_A(A) - \mu_{A|C}(C) \right) - \delta_{A|C}^{(t)}(C),$$

and similarly for $B$, we note that $\mu_{A|C}^{(t)}$, $\mu_{B|C}^{(t)}$, and hence $\delta_{A|C}^{(t)}$, $\delta_{B|C}^{(t)}$, are $\mathcal{F}_{t-1}$-measurable.

Under $H_0$, $A \perp\!\!\!\perp B \mid C$. Moreover,

$$\bar{\mathbb{E}}_{H_0} \left[ \phi_A(A) - \mu_{A|C}(C) \mid C \right] = 0, \qquad \bar{\mathbb{E}}_{H_0} \left[ \phi_B(B) - \mu_{B|C}(C) \mid C \right] = 0.$$

Therefore, after taking the conditional expectation given $C$, all terms containing at least one population residual vanish. The only remaining term is the product of the regression errors. Hence

$$\bar{\mathbb{E}}_{H_0} \left[ \psi^{(t)}(Y) \mid \mathcal{F}_{t-1} \right] = \bar{\mathbb{E}}_C \left[ \delta_{A|C}^{(t)}(C) \otimes \delta_{B|C}^{(t)}(C) \otimes \phi_C(C) \right].$$

Consequently,

$$\mu_t = \frac{1}{S_n^{(t-1)} + \varepsilon} \left\langle \frac{1}{n} \sum_{i=1}^n \psi^{(t)}(x_i), \bar{\mathbb{E}}_C \left[ \delta_{A|C}^{(t)}(C) \otimes \delta_{B|C}^{(t)}(C) \otimes \phi_C(C) \right] \right\rangle_{\mathcal{H}}. \tag{18}$$

Applying Cauchy–Schwarz and the triangle inequality,

$$|\mu_t| \leq \frac{1}{S_n^{(t-1)} + \varepsilon} \left\| \frac{1}{n} \sum_{i=1}^{n} \psi^{(t)}(x_i) \right\|_{\mathcal{H}} \left\| \bar{\mathbb{E}}_C \left[ \delta_{A|C}^{(t)}(C) \otimes \delta_{B|C}^{(t)}(C) \otimes \phi_C(C) \right] \right\|_{\mathcal{H}}$$

$$\leq \frac{1}{S_n^{(t-1)} + \varepsilon} \left( \frac{1}{n} \sum_{i=1}^{n} \|\psi^{(t)}(x_i)\|_{\mathcal{H}} \right) \bar{\mathbb{E}}_C \left[ \|\delta_{A|C}^{(t)}(C)\| \|\delta_{B|C}^{(t)}(C)\| \|\phi_C(C)\| \right].$$

Since

$$\|\psi^{(t)}(x_i)\|_{\mathcal{H}} = \sqrt{h^{(t)}(x_i, x_i)} \leq \sqrt{\kappa},$$

and $\sup_c \|\phi_C(c)\| \leq 1$, we have

$$\bar{\mathbb{E}}_C \left[ \|\delta_{A|C}^{(t)}(C)\| \|\delta_{B|C}^{(t)}(C)\| \|\phi_C(C)\| \right] \leq \|\delta_{A|C}^{(t)}\| \|\delta_{B|C}^{(t)}\|.$$

Here we used

$$\|\delta_{A|C}^{(t)}(C)\| \leq \|\delta_{A|C}^{(t)}\| \|\phi_C(C)\| \leq \|\delta_{A|C}^{(t)}\|,$$

and similarly for $\delta_{B|C}^{(t)}$. Therefore,

$$|\mu_t| \leq \frac{\sqrt{\kappa}}{S_n^{(t-1)} + \varepsilon} \|\delta_{A|C}^{(t)}\| \|\delta_{B|C}^{(t)}\|.$$

Since $S_n^{(t-1)}$ is the V-statistic estimator of a squared RKHS norm, we have $S_n^{(t-1)} \geq 0$. Hence $S_n^{(t-1)} + \varepsilon \geq \varepsilon$. It follows that

$$|\mu_t| \leq \frac{\sqrt{\kappa}}{\varepsilon} \|\delta_{A|C}^{(t)}\| \|\delta_{B|C}^{(t)}\|.$$

$\square$

**Lemma A.4** (Bound on the Variance Mismatch)**.** *Assume the kernel is bounded such that $\sup_x h^{(t)}(x,x) \leq \kappa$. The difference between the $\sigma_t^2$ and $\widehat{\sigma}_t^2$ is bounded by*

$$|\sigma_t^2 - \widehat{\sigma}_t^2| \leq \frac{2\kappa^2}{b\varepsilon^2}. \tag{19}$$

*Proof.* Recall that the raw betting statistic can be written as

$$V_t^{\mathrm{raw}} = \frac{1}{b} \sum_{j=1}^{b} g^{(t)}(Y_j).$$

Under the null and conditional on $\mathcal{F}_{t-1}$, the variables $\{g^{(t)}(Y_j)\}_{j=1}^{b}$ are i.i.d. Hence the conditional variance of $V_t^{\mathrm{raw}}$ is

$$\sigma_t^2 = \frac{1}{b} \overline{\mathrm{Var}}_{H_0}\left( g^{(t)}(Y) \right) = \frac{1}{b} \bar{\mathbb{E}}_{H_0} \left[ (g^{(t)}(Y))^2 \right] - \frac{1}{b} \mu_t^2,$$

where $Y$ is a fresh null draw and

$$\mu_t = \bar{\mathbb{E}}_{H_0} \left[ g^{(t)}(Y) \right].$$

The empirical variance-scale estimator used for calibration is

$$\widehat{\sigma}_t^2 = \frac{1}{b} \left( \frac{1}{b} \sum_{j=1}^{b} (g^{(t)}(v_j))^2 \right) = \frac{1}{b^2} \sum_{j=1}^{b} (g^{(t)}(v_j))^2,$$

where $\{v_j\}_{j=1}^{b}$ are validation samples independent of the training data used to construct $g^{(t)}$.

Therefore,

$$|\sigma_t^2 - \widehat{\sigma}_t^2| \leq |\sigma_t^2| + |\widehat{\sigma}_t^2|$$

$$\leq \frac{1}{b}\left\{\bar{\mathbb{E}}_{H_0}\left[(g^{(t)}(Y))^2\right] + \frac{1}{b}\sum_{j=1}^{b}(g^{(t)}(v_j))^2\right\}.$$

Moreover, by the boundedness assumption, $\|\psi^{(t)}(x_i)\|_{\mathcal{H}} = \sqrt{h^{(t)}(x_i, x_i)} \leq \sqrt{\kappa}$. Hence

$$|g^{(t)}(Y)| = \left|\frac{1}{n(S_n(\mathcal{X}_{t-1}^{\mathrm{tr}}) + \varepsilon)}\sum_{i=1}^{n}h^{(t)}(x_i, Y)\right|$$

$$\leq \frac{1}{n\varepsilon}\sum_{i=1}^{n}|h^{(t)}(x_i, Y)|$$

$$= \frac{1}{n\varepsilon}\sum_{i=1}^{n}\left|\left\langle\psi^{(t)}(x_i), \psi^{(t)}(Y)\right\rangle_{\mathcal{H}}\right|$$

$$\leq \frac{1}{n\varepsilon}\sum_{i=1}^{n}\|\psi^{(t)}(x_i)\|_{\mathcal{H}}\|\psi^{(t)}(Y)\|_{\mathcal{H}}$$

$$\leq \frac{\kappa}{\varepsilon}.$$

Similarly, $|g^{(t)}(v_j)| \leq \frac{\kappa}{\varepsilon}$ for each $j \in \{0, \ldots, b\}$. Therefore,

$$|\sigma_t^2 - \widehat{\sigma}_t^2| \leq \frac{2\kappa^2}{b\varepsilon^2}.$$

This completes the proof. $\qquad\square$

We now return to Term (II). By the shift sensitivity bound,

$$\text{Term (II)} \leq \max\{\mu_t + \Lambda(\sigma_t)(\sigma_t - \widehat{\sigma}_t), 0\}.$$

Hence

$$\text{Term (II)} \leq |\mu_t + \Lambda(\sigma_t)(\sigma_t - \widehat{\sigma}_t)|$$

$$\leq |\mu_t| + \Lambda(\sigma_t)|\sigma_t - \widehat{\sigma}_t|$$

$$= |\mu_t| + \Lambda(\sigma_t)\frac{|\sigma_t^2 - \widehat{\sigma}_t^2|}{\sigma_t + \widehat{\sigma}_t}$$

$$\leq |\mu_t| + \frac{\Lambda(\sigma_t)}{\sigma_t}|\sigma_t^2 - \widehat{\sigma}_t^2|.$$

Recall that $\Lambda(\sigma_t) = \frac{\phi(\xi_{\sigma_t})}{\Phi(-\xi_{\sigma_t})}$, where $\xi_{\sigma_t}$ solves $\phi(\xi) - \xi\Phi(-\xi) = \frac{1}{\sigma_t}$. Thus

$$\frac{\Lambda(\sigma_t)}{\sigma_t} = \frac{\phi(\xi_{\sigma_t})^2}{\Phi(-\xi_{\sigma_t})} - \xi_{\sigma_t}\phi(\xi_{\sigma_t}),$$

which is uniformly bounded above by a finite constant $C_2$. Combining this with the mean and variance mismatch bounds gives

$$\text{Term (II)} \leq |\mu_t| + C_2|\sigma_t^2 - \widehat{\sigma}_t^2|$$

$$\leq \frac{\sqrt{\kappa}}{\varepsilon}\|\delta_{A|C}^{(t)}\|\|\delta_{B|C}^{(t)}\| + \frac{2C_2\kappa^2}{b\varepsilon^2}.$$

## A.4. Finite-Sample Type I Error Bound

In this section, we translate the one-step drift bound into a finite-sample bound on the Type I error under $H_0$. Combining the Gaussian approximation bound for Term (I) with the calibration mismatch bound for Term (II), we obtain

$$\delta_t = \text{Term (I)} + \text{Term (II)} \leq \frac{C_1 \rho}{b\,\varepsilon} + \frac{\sqrt{\kappa}}{\varepsilon}\|\delta_{A|C}^{(t)}\|\|\delta_{B|C}^{(t)}\| + \frac{2C_2 \kappa^2}{b\,\varepsilon^2}. \tag{20}$$

Let $U_t$ denote the right-hand side of Equation (20).

**Theorem A.5** (Finite-Sample Type I Error). *Let $\alpha \in (0,1)$. Suppose that under $H_0$, for every $t \leq T$,*

$$\delta_t = \mathbb{E}_{H_0}[V_t \mid \mathcal{F}_{t-1}] \leq U_t,$$

*where*

$$U_t := \frac{C_1 \rho}{b\,\varepsilon} + \frac{\sqrt{\kappa}}{\varepsilon}\|\delta_{A|C}^{(t)}\|\|\delta_{B|C}^{(t)}\| + \frac{2C_2 \kappa^2}{b\,\varepsilon^2}.$$

*Then*

$$\Pr_{H_0}\left(\exists t \leq T : W_t \geq \frac{1}{\alpha}\right) \leq \alpha \exp\left(\sum_{t=1}^{T} \lambda_t U_t\right).$$

*Proof.* Define the compensated process

$$\widetilde{W}_t := \frac{W_t}{\prod_{i=1}^{t}(1 + \lambda_i U_i)}.$$

Since $V_t \geq -1$ and $\lambda_t \in (0,1)$, the wealth process is nonnegative. Moreover, by the assumed drift bound,

$$\begin{aligned}
\mathbb{E}_{H_0}[\widetilde{W}_t \mid \mathcal{F}_{t-1}] &= \frac{\mathbb{E}_{H_0}[W_t \mid \mathcal{F}_{t-1}]}{\prod_{i=1}^{t}(1 + \lambda_i U_i)} \\
&= \frac{W_{t-1}\,\mathbb{E}_{H_0}[1 + \lambda_t V_t \mid \mathcal{F}_{t-1}]}{\prod_{i=1}^{t}(1 + \lambda_i U_i)} \\
&= \frac{W_{t-1}(1 + \lambda_t \delta_t)}{\prod_{i=1}^{t}(1 + \lambda_i U_i)} \\
&\leq \frac{W_{t-1}(1 + \lambda_t U_t)}{\prod_{i=1}^{t}(1 + \lambda_i U_i)} = \widetilde{W}_{t-1}.
\end{aligned}$$

Thus $(\widetilde{W}_t)_{t=0}^{T}$ is a nonnegative supermartingale with $\widetilde{W}_0 = 1$.

If $W_t \geq 1/\alpha$, then

$$\widetilde{W}_t = \frac{W_t}{\prod_{i=1}^{t}(1 + \lambda_i U_i)} \geq \frac{1}{\alpha \prod_{i=1}^{t}(1 + \lambda_i U_i)}.$$

Equivalently,

$$W_t = \widetilde{W}_t \prod_{i=1}^{t}(1 + \lambda_i U_i).$$

Since

$$\prod_{i=1}^{t}(1 + \lambda_i U_i) \leq \exp\left(\sum_{i=1}^{t} \lambda_i U_i\right) \leq \exp\left(\sum_{i=1}^{T} \lambda_i U_i\right),$$

we have

$$W_t \leq \widetilde{W}_t \exp\left(\sum_{i=1}^{T} \lambda_i U_i\right).$$

Therefore,

$$W_t \geq \frac{1}{\alpha} \quad \Longrightarrow \quad \widetilde{W}_t \geq \frac{\exp\left(-\sum_{i=1}^{T} \lambda_i U_i\right)}{\alpha}.$$

By Ville's inequality,

$$
\Pr_{H_0}\left(\exists t \leq T : W_t \geq \frac{1}{\alpha}\right) \leq \Pr_{H_0}\left(\exists t \leq T : \widetilde{W}_t \geq \frac{\exp\left(-\sum_{i=1}^{T} \lambda_i U_i\right)}{\alpha}\right)
$$

$$
\leq \alpha \exp\left(\sum_{i=1}^{T} \lambda_i U_i\right).
$$

This proves the claim. $\qquad\square$

## B. Implementation Details

### B.1. Datasets

**Synthetic neural data.** Following Pogodin et al. (2024), we evaluate the tests on high-dimensional synthetic neural data generated with RatInABox (George et al., 2024). The setup follows the implementation of Pogodin et al. (2024), available at `github.com/romanpogodin/kernel-ci-testing`, and is based on the head-direction and conjunctive-cell models from the RatInABox conjunctive-cells demo.

We take $A$ to be the activity of 100 head-direction cells and $B$ to be the activity of 100 conjunctive cells. The conditioning variable $C$ consists of the two-dimensional head-direction vector and the two-dimensional position of the agent. The resulting testing question is whether the head-direction-cell activity $A$ contains information about the conjunctive-cell activity $B$ beyond what is explained by head direction and position $C$. As in Pogodin et al. (2024), the activity is subsampled based on the noise autocorrelation to obtain approximately i.i.d. observations.

**dSprites.** Inspired by Zhang et al. (2025), we construct a conditional independence task from dSprites (Matthey et al., 2017). Each observation is a $64 \times 64$ image containing a single object. We fix the object scale and use the shape label as $A$. The variable $B$ is the full image. The conditioning variable $C$ is a $28 \times 28$ crop derived from $B$, padded or placed on a fixed canvas.

The two testing regimes differ in how the crop is chosen. In the Type I setting, $C$ is obtained from the tight bounding box of the object and hence contains the full object. In this case, once $C$ is given, $B$ should not provide substantial additional information about the shape, so $A \perp\!\!\!\perp B \mid C$ should hold. In the Type II setting, the crop is randomly shifted away from the object bounding box, so $C$ typically contains only partial object information. Then $B$ contains additional information about $A$ beyond $C$, and the conditional independence null is false.

**Car insurance data.** Following the experiments of Polo et al. (2023); Pogodin et al. (2024), we also evaluate the methods on a car insurance dataset originally collected by Angwin et al. (2017) from four US states and multiple insurance companies.[3] The dataset contains three variables: car insurance price $A$, minority neighborhood indicator $B$, and driver risk $C$. The minority neighborhood indicator is defined as more than $66\%$ non-white in California and Texas, and more than $50\%$ non-white in Missouri and Illinois. The risk variable $C$ is constructed from driver-related risk factors, with additional equalization of driver characteristics; see Angwin et al. (2017) for details. The resulting conditional independence question is whether insurance price remains associated with the minority-neighborhood indicator after conditioning on driver risk.

### B.2. Kernel Choices in SKCI

SKCI employs separate kernels for variables $A$, $B$, and $C$. Unless otherwise stated, all kernels are fixed throughout a run and updated only through improved conditional mean estimates as more data become available.

For the kernels on $A$ and $B$, we use an RBF kernel when the variables are low-dimensional. The bandwidth is fixed to 1 for synthetic datasets and initialized using a variance heuristic for the car insurance data. The conditioning kernels $k_{C \to A}$ and $k_{C \to B}$ are selected by leave-one-out kernel ridge regression with per-dimension lengthscales, with early stopping based on a validation set. We use the same kernel family for conditional mean estimation and for the kernel on $C$.

For Gaussian data, conditional means are estimated using an MLP feature map with layer sizes $[128, 64, 32, 8]$, followed by

---

[3]Data available from `https://projects.propublica.org/graphics/carinsurance`.

an RBF kernel.

For dSprites, we use convolutional feature maps before applying RBF kernels. The full image $B$ is encoded by a pretrained autoencoder for $64 \times 64$ single-channel images, with hidden channel widths $[32, 64, 128]$ and a 16-dimensional latent representation. The autoencoder is trained on the same training samples used for the corresponding regression step and is then frozen during kernel fitting. Kernels involving $B$ are computed on this latent representation, with the RBF bandwidth initialized by the variance heuristic. The conditioning variable $C$ is a $28 \times 28$ single-channel crop. For the kernel on $C$, and for the conditional mean maps $C \to A$ and $C \to B$, we use trainable CNN feature maps with hidden channel widths $[32, 64, 128]$ and a fully connected hidden layer of width 128. The output dimensions are 8 for the kernel on $C$, 1 for the $C \to A$ map, and 8 for the $C \to B$ map.

For the remaining datasets, we apply RBF, Kronecker, or linear kernels directly, following the dataset-specific kernel choices used in Pogodin et al. (2024); He et al. (2025).

*Table 1.* Kernel choices across datasets.

| Dataset | $k_A$ / $k_B$ | $k_C$ / $k_{C \to A}$ / $k_{C \to B}$ | Notes |
|---|---|---|---|
| Gaussian | RBF | RBF + MLP | 1-dimensional A, B |
| Hardness (1D/3D) | RBF | RBF | 1-dimensional A, B |
| RatInABox | Linear | RBF | 100-dimensional $A, B$ |
| dSprites | RBF / RBF + AutoEncoder | RBF + CNN | 1-dimensional A; image B, C |
| Car Insurance | RBF / Kronecker | RBF | 1-dimensional A; categorical $B$ |

## B.3. Conditional Mean Estimation Modes

We evaluate SKCI under three estimation regimes:

- **Oracle Mode**: The true conditional mean $\mathbb{E}[A \mid C]$ is provided for $C \to A$ regression task as the target.

- **Pretrained Mode**: Conditional mean estimators are trained using 3,000 auxiliary samples independent of the testing stream.

- **Online Mode**: Conditional mean estimators are updated sequentially using the accumulated training history.

## B.4. Baselines and Fair Comparison

We compare SKCI against e-CRT (Shaer et al., 2023), DAVT (Pandeva et al., 2024b) and EC2ST (Pandeva et al., 2024a).

For baseline methods, all regressors assume a Gaussian conditional model $A \mid C \sim \mathcal{N}(\mu(C), \sigma^2)$. In Oracle mode, both $\mu(C)$ and $\sigma^2$ are provided; otherwise, they are estimated from data. Hyperparameters are selected using an additional set of 1,000 samples, with the search space shown in Table 2, via 5-fold cross-validation over 5 independent data seeds. Final architecture choices and tuned hyperparameters for each dataset (Gaussian, Hardness 1D/3D, RatInABox, dSprites, and Car Insurance) are reported in Table 3.

*Table 2.* Hyperparameter search space for baseline models.

| Parameter | Values |
|---|---|
| Learning rate | $\{0.1, 0.01, 0.001, 0.0001\}$ |
| Weight decay | $\{0, 10^{-5}, 10^{-4}\}$ |
| Hidden channels | $\{[32], [32, 64], [32, 64, 128], [32, 64, 128, 256]\}$ |
| Hidden layers | $\{[32], [64], [128], [64, 32], [128, 64], [128, 64, 32]\}$ |

In addition to conditional mean estimation, baseline models also rely on discriminator or regression models whose hyperparameters are tuned separately. For DAVT, the discriminator distinguishing $Z$ from $\tilde{Z}$ selected based on validation loss aligned with the wealth objective $g(Z) - g(\tilde{Z})$. For e-CRT, the discriminator uses regression loss with $(A, C)$ as input and $B$ as target. The corresponding search spaces are the same as the search space shown in Table 2, while DAVT

*Table 3.* Model configurations for conditional mean estimation.

| Dataset | Model | Hidden channels | Hidden dims | LR | Weight decay |
|---|---|---|---|---|---|
| Gaussian | MLP | – | [32] | 0.01 | 0 |
| CI Hardness (1D) | MLP | – | [128] | 0.01 | 0 |
| CI Hardness (3D, shared) | MLP | – | [64,32] | 0.01 | $10^{-4}$ |
| CI Hardness (3D, sep.) | MLP | – | [32] | 0.01 | 0 |
| RatInABox | MLP | – | [128,64] | 0.01 | 0 |
| dSprites | CNN | [32, 64] | [32] | 0.001 | 0 |
| Car insurance | MLP | – | [128,64] | 0.01 | 0 |

additionally searches learning rates in $\{0.01, 0.005, 0.001, 0.0005, 0.0001\}$. Dataset-specific configures for discriminator architectures and optimization parameters are reported in Table 4.

For the image-based dSprites experiment, we use CNN-based models. For DAVT and EC2ST, the network processing $B$ uses hidden channel widths $[32, 64, 128]$, while the network processing the conditioning information uses hidden channel widths $[32, 64, 128, 256]$. For e-CRT, the model uses an encoder–decoder architecture, with encoder hidden channel widths $[32, 64]$ and decoder hidden channel widths $[64, 32]$.

*Table 4.* Discriminator configurations for each dataset.

| Dataset | Method | Hidden dims | LR | Weight decay |
|---|---|---|---|---|
| Gaussian | DAVT | [128] | 0.0005 | 0 |
| CI Hardness (1D) | DAVT | [128] | 0.001 | $10^{-5}$ |
| CI Hardness (3D, shared) | DAVT | [128] | 0.005 | 0 |
| CI Hardness (3D, separate) | DAVT | [128] | 0.005 | 0 |
| RatInABox | DAVT | [128] | 0.0005 | $10^{-5}$ |
| dSprites | DAVT | [64] | 0.001 | $10^{-4}$ |
| Car insurance | DAVT | [128] | 0.0005 | $10^{-5}$ |
| Gaussian | e-CRT | [128,64] | 0.01 | $10^{-5}$ |
| CI Hardness (1D) | e-CRT | [128,64,32] | 0.01 | $10^{-4}$ |
| CI Hardness (3D, shared) | e-CRT | [128,64,32] | 0.01 | $10^{-4}$ |
| CI Hardness (3D, separate) | e-CRT | [128,64] | 0.01 | $10^{-5}$ |
| RatInABox | e-CRT | [128] | 0.01 | $10^{-4}$ |
| dSprites | e-CRT | [32] | 0.01 | $10^{-4}$ |
| Car insurance | e-CRT | [128] | 0.01 | $10^{-4}$ |
| Gaussian | EC2ST | [128] | 0.0005 | 0 |
| CI Hardness (1D) | EC2ST | [128] | 0.001 | $10^{-5}$ |
| CI Hardness (3D, shared) | EC2ST | [128] | 0.005 | 0 |
| CI Hardness (3D, separate) | EC2ST | [128] | 0.005 | 0 |
| RatInABox | EC2ST | [128] | 0.0005 | $10^{-5}$ |
| dSprites | EC2ST | [64] | 0.001 | $10^{-4}$ |
| Car insurance | EC2ST | [128] | 0.0005 | $10^{-5}$ |

## C. Additional Results

### C.1. Results for Oracle and Pretrained Mode

Figures 6 to 9 give emiprical results for the synthetic problems in Oracle and Pretrained mode.

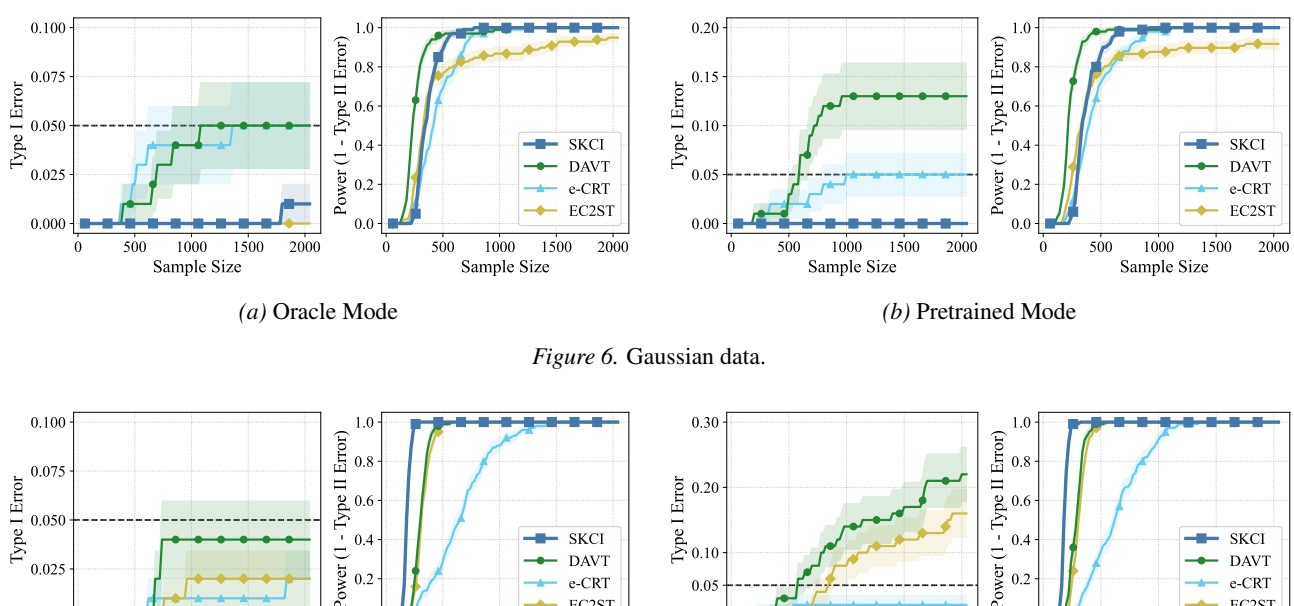

*(a)* Oracle Mode                  *(b)* Pretrained Mode

*Figure 6.* Gaussian data.

*(a)* Oracle Mode                  *(b)* Pretrained Mode

*Figure 7.* CI Hardness data (1D).

## C.2. Long Time Horizon Experiments

Figure 11 reports long-horizon runs under the null, in which we keep the hyperparameters fixed and only increase the time horizon $T$. The cumulative number of rejections stabilizes as the sample size increases. In both settings, most rejections occur early in the online sequence, and the reject count remains nearly constant after a moderate number of samples. This suggests that, when the conditional mean estimates are sufficiently accurate, the rejection count does not necessarily continue to grow with the time horizon.

## C.3. Ablation

**Batch Size** Figure 12a shows that smaller batch sizes accelerate power accumulation but may slightly increase variance in early rounds. The default choice $b = 20$ offers a good balance between responsiveness and stability.

**Regularization Parameter** Figure 12b illustrates the effect of the denominator regularization $\varepsilon$. Too small values may lead to unstable payoffs, while overly large values reduce power. Across experiments, we fix $\varepsilon = 10^{-6}$ which provides robust performance.

**Shift Parameter** Figure 12c examines the effect of using a more conservative shift parameter $\gamma$. Since the payoff is nonincreasing in $\gamma$, increasing $\gamma$ reduces the size of the wealth update and can therefore make the test more conservative. We compare the default choice $\hat{\gamma}_t$ with the enlarged value $\hat{\gamma}_t + 0.1$. The enlarged choice reduces Type I error while leaving power essentially unchanged in this experiment, suggesting that the procedure is not highly sensitive to small conservative perturbations of $\hat{\gamma}_t$.

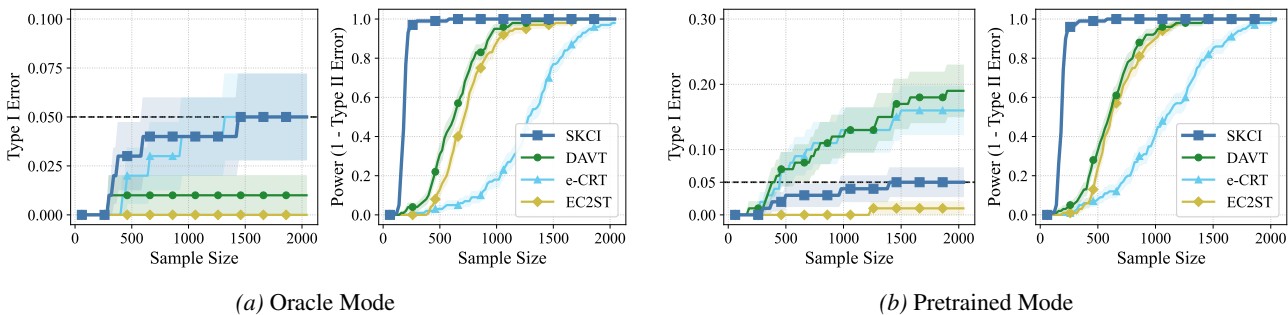

*(a)* Oracle Mode  *(b)* Pretrained Mode

*Figure 8.* CI Hardness data (3D, shared coordinate).

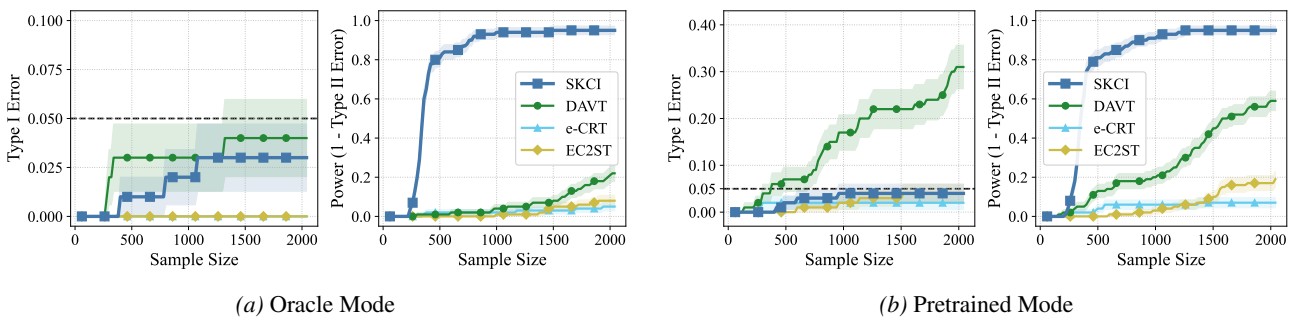

*(a)* Oracle Mode  *(b)* Pretrained Mode

*Figure 9.* CI Hardness data (3D, separate coordinate).

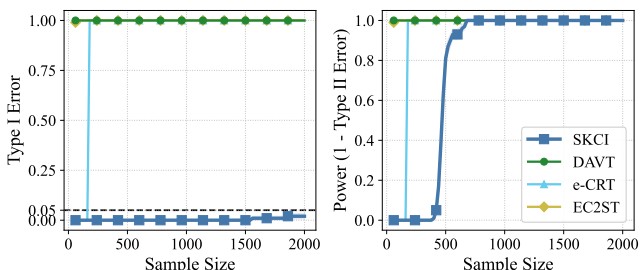

*Figure 10.* Synthetic neural data, Pretrained Mode.

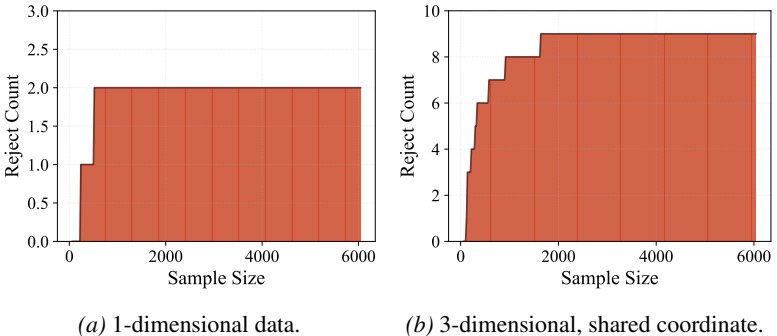

*(a)* 1-dimensional data.  *(b)* 3-dimensional, shared coordinate.

*Figure 11.* Long horizon experiments on CI hardness data under the null, Online mode, 100 runs.

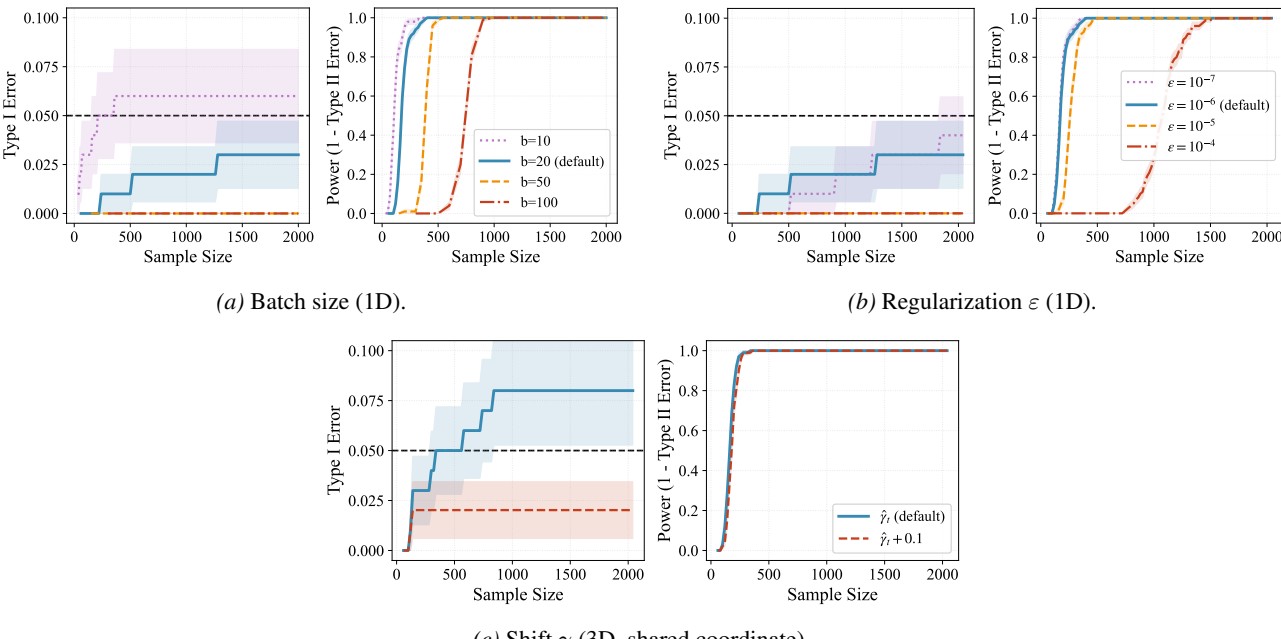

*(a)* Batch size (1D).

*(b)* Regularization $\varepsilon$ (1D).

*(c)* Shift $\gamma$ (3D, shared coordinate).

*Figure 12.* Ablation studies for SKCI on CI-hardness data, Online mode. Panels (a) and (b) show the effect of varying the batch size and denominator regularization parameter $\varepsilon$ on the one-dimensional setting. Panel (c) studies a conservative perturbation of the shift parameter on the three-dimensional shared-coordinate setting, comparing the default $\hat{\gamma}_t$ with $\hat{\gamma}_t + 0.1$.

