# OpenReview forum: "Sequential Kernel-based Conditional Independence Testing via Adaptive Betting"
_ICML.cc/2026/Conference — ICML 2026 regular_

### Official Review · Reviewer_Lzgo · 2026-03-04

**Soundness:** 3
**Presentation:** 4
**Significance:** 3
**Originality:** 3
**Overall Recommendation:** 5
**Confidence:** 3

**Summary:**

The paper introduces Sequential Kernel-based Conditional independence (SKCI), a robust framework for anytime-valid testing in streaming data settings. Unlike existing work that requires a ground-truth distributions ("Model-X"), SKCI utilizes a truncate-and-shift calibration and adaptive kernel normalization within a testng-by-betting paradigm for Type I error control.

**Compliance With Llm Reviewing Policy:**

Affirmed.

**Final Justification:**

The rebuttal comment clarified my concern, thus I maintain the original high score.

**Key Questions For Authors:**

Please provide a point-by-point response to the concerns listed above.

**Limitations:**

yes

**Strengths And Weaknesses:**

The paper is well-written and an enjoyable read. Note that the following represents my initial impression of the paper, and I am open to discussion. I welcome any corrections to my potential  misunderstandings.

## Strengths
- **Oracle-free CI testing**: SKCI runs on real-world datasets without "Model-X" ground-truth knowledge.
- **Anytime-valid reliability of SKCI** that provides statistically sound results under streaming data collection, optional stopping, and multiple testing.
- **High statistical power**: SKCI achieves rapid power accumulation, tested over 100 independent runs. Plots also have standard error, allowing readers to assess stability and reproducibility of the performance (which is still relatively rare in machine learning literature - I highly appreciate it)
- Novel **self-normalized payoff** $U_{n, b}/(S_n+\epsilon)$ and **Truncate-and-shift calibration** to control Type I error rate.
- **Theoretical foundation.** The authors provide a robust anytime validity bound.

## Weaknesses
**Anytime-valid claims requires an exact nonnegative supermartingale under $H_0$**: however, as the paper acknowledges, the wealth process is not an exact supermartingale outside Model-X regime, and instead provides a finite-horizon bound with an inflation factor $\exp(\sum_{t<T}\lambda_t U_t)$. The asymptotic gap term does not decay with time in their decomposition, which could be problematic because online settings often mean potentially long streams. Thus a clearer statement of what is guaranteed outside Model-X with empirical stress tests over much longer horizons (with fixed batch-size $b$) will strengthen the paper further.

**Analysis is currently limited in i.i.d. observations**. The introduction motivates settings with distribution shift and other real-world complexities. But if the intended use includes non-stationality, it is beyond the current theory's scope and thus the paper should be explicit that the theoretical analysis is i.i.d.-based. Having said that, i.i.d. setting is a reasonable and defensible first step beyond the Model-X regime.

---

> ### Author Rebuttal · Authors · 2026-03-31
>
> We thank the reviewer for the careful reading and for the positive assessment of the paper’s motivation, methodological, empirical, and theoretical contributions. We are encouraged that the reviewer views the setting beyond Model-X as meaningful and appreciates the effort to provide a principled sequential CI testing framework. We address the two main concerns below.
>
> ### Clarifying the guarantee outside Model-X and long-horizon behavior
>
> We thank the reviewer for bringing up this important concern.
>
> We agree that the wealth process is not an exact nonnegative supermartingale under $H_0$. Our result therefore provides a finite-horizon, approximation-aware validity guarantee, where the type I error is controlled up to an explicit drift term $U_t$ depending on the estimation errors and asymptotic gap. We will revise the presentation to make this distinction explicit and avoid overstating “anytime validity” beyond Model-X.
>
> Regarding long-horizon behavior, two effects are involved. Estimation bias can decrease over time as conditional models are trained with more data. In contrast, the asymptotic gap in our bound is governed by the batch size b and does not vanish with t when b is fixed. We will revise the paper to make this distinction explicit.
>
> Empirically, we examined longer horizons by keeping the batch size $b=20$ and extending the testing sequence (e.g., T=300). We observe that the Type I error stabilizes (rather than growing with time) after an initial phase.
>
> Type I Error: CI hardness data, d=1, online mode
> |data size| 1000 | 2000 | 3000 | 4000 | 5000 | 6000 |
> |:-|:-|:-|:-|:-|:-|:-|
> | SKCI     |  0.02±0.01 | 0.02±0.01 | 0.02±0.01 | 0.02±0.01 | 0.02±0.01 | 0.02±0.01 |
>
> Type I Error: CI hardness data, d=3, shared coordinate, online mode
> |data size| 1000 | 2000 | 3000 | 4000 | 5000 |6000|
> |:-|:-|:-|:-|:-|:-|:-|
> | SKCI| 0.08±0.03 | 0.09±0.03 | 0.09±0.03 | 0.09±0.03 | 0.09±0.03 | 0.09±0.03 |
>
> We also demonstrate a practical way to improve robustness via more conservative calibration of $\gamma_t$. Increasing $\gamma_t$ induces a negative drift in the payoff, which helps reduce the effective drift term. In particular, Lemma 4.2 shows that the leading asymptotic gap scales as $\rho/\sqrt{b}$; increasing $\gamma_t$ at a comparable scale partially offsets this component. While this does not eliminate the gap entirely, it provides a practical robustness–power tradeoff.
>
>
> In the following experiment, increasing $\gamma_t$ by 0.1 improves Type I error control (e.g., from ~0.08 to ~0.02), at the cost of slower power accumulation:
>
>
> Type I Error: CI hardness data, d=3, shared coordinate, online mode
> | Vt type | 200 | 400 |800 | 1000 ... 2000 |
> |:-|:-|:-|:-|:-|
> | $V_t$ (ours) | 0.03±0.02 | 0.05±0.02 | 0.07±0.03 | 0.08±0.03 |
> | $V_t$ (increase $\hat \gamma_t$ by 0.1) | 0.02±0.01 | 0.02±0.01 | 0.02±0.01 | 0.02±0.01 |
>
> Power:  CI hardness data, d=3, shared coordinate, online mode
> | Vt type | 200 | 400 ... 2000 |
> |:-|:-|:-|
> | $V_t$ (ours) | 0.83±0.04 | 1.00±0.00 |
> | $V_t$ (increase $\hat \gamma_t$ by 0.1) | 0.72±0.05 | 1.00±0.00 | 1.00±0.00 |
>
>
> ### Scope of the current theory
> We fully agree that the present analysis is based on i.i.d./stationary sampling. Although the paper is motivated by streaming and optional stopping, the current theory does not cover general non-stationary or drifting data streams. We will make this scope explicit and position the paper as a first step beyond Model-X under stationary sequential monitoring.
>
> ### Planned revisions
>
> We will revise the paper to:
> - clarify that the result is finite-horizon sequential validity, rather than exact infinite-horizon anytime validity;
> - add longer-horizon experiments, and systematically analyze the effect of $\gamma$ on validity and power;
> - and make explicit that the current theory assumes stationary/i.i.d. data.
>
>
> ---
> We greatly appreciate the reviewer’s thoughtful and encouraging comments. It is very helpful to know that the motivation, methodology, and empirical results are well received. We will incorporate the reviewer’s suggestions to further improve the paper.

---

> > ### Author Rebuttal · Reviewer_Lzgo · 2026-04-02
> >
> > Thank you for the clarification—this addresses my concerns regarding the scope of the guarantees and the i.i.d. setting, and I appreciate the additional empirical results on longer horizons. Given this, I view the limitations as well-understood and not grounds for rejection, and I will maintain my positive score.

---

### Official Review · Reviewer_YLnf · 2026-03-11

**Soundness:** 3
**Presentation:** 3
**Significance:** 3
**Originality:** 3
**Overall Recommendation:** 4
**Confidence:** 4

**Summary:**

The paper introduces a sequential conditional independence (CI) test by combining a conditional kernel-based test with adaptive betting methods. Compared with Model-X-based sequential CI tests, the proposed approach relaxes the assumption that the conditional distribution is known. The methodology represents a novel integration of existing techniques, and the paper provides theoretical justifications for the proposed procedures. However, a more careful discussion of conditional mean estimation and kernel selection is needed, particularly in high-dimensional settings. In addition, although the data are described as sequentially observed, they appear to be treated as a stationary sequence in which the entire sample comes either from the null or from the alternative.

**Compliance With Llm Reviewing Policy:**

Affirmed.

**Final Justification:**

The authors have clarified most of the concerns during the rebuttal. They also noted that the paper does not address conditional independence testing in high-dimensional settings. However, the numerical studies include experiments with high-dimensional variables, which creates some inconsistency with the scope of the theoretical results. Despite these limitations, and given the limited time during the rebuttal period, I will retain my positive recommendation.

**Key Questions For Authors:**

1. The conditional mean estimation is essential in a conditional independent test, especially when the conditional variable $C$ is a high-dimensional random variable. How does the conditional mean estimation affect the theoretical and empirical results in the paper?
2. In the synthetic neural data example, both $A$ and $B$ are high-dimensional. How does the high-dimensionality of $A$ and $B$ affect the proposed kernel-based conditional test for high-dimensional random variables? In addition, the choice of kernel and hyperparameters in the procedure should be considered carefully. There are studies, such as He et al. (2023) (https://www.jstor.org/stable/27249945?seq=1), that have considered kernel-based tests for high-dimensional random variables.
3. Can the proposed framework handle non-stationary sequential data, where the distribution may change over time?
4. How does the method compare empirically with recent Model-X sequential CI tests when the Model-X assumption approximately holds?

**Limitations:**

Yes.

**Strengths And Weaknesses:**

Strengths:

1. The conditional independence testing is a fundamental and  important problem.
2. The paper employs a novel combination of adaptive betting and conditional kernel testing.
2. The general procedure and implementation are well-introduced.
3. Some theoretical guarantees are provided for the proposed procedure.

Weakness:
1. Although the setting is described as sequential, the observations appear to be treated effectively as independent draws from a stationary distribution, where the entire sequence is assumed to come either from the null or from the alternative. The implications for genuinely non-stationary or adaptive data streams are not clearly discussed.
2. The method relies on estimating conditional expectations in the construction of the kernel-based statistic. The paper provides limited discussion on the practical estimation of these quantities, particularly in high-dimensional settings, where estimation errors may significantly affect the validity or power of the test.
3. The performance of kernel-based tests is sensitive to the choice of kernel and its hyperparameters, but the paper provides little guidance on how these should be selected in practice.
4. Some steps of the algorithm, including the estimation procedure used within each round and how hyperparameters are tuned, could be explained more clearly.

---

> ### Author Rebuttal · Authors · 2026-03-31
>
> Thanks for the constructive and thoughtful comments; we'll clarify all points in revision.
> ### Clarifying the sequential setting and scope
> “Sequential” in our paper indeed refers to sequential monitoring under optional stopping, where data arrive over time but are assumed iid. Like the vast majority of work in ML and statistics, the current framework does not address non-stationarity. Extending the framework to non-stationary settings would be very interesting, but is outside the scope of the current paper.
>
> ### Role of conditional mean estimation in high-dimensional settings
> We agree this is central to CI testing beyond Model-X and is a primary motivation of our work. Estimation quality directly affects both validity (via Type I inflation) and power (via residual dependence). Our theory makes this explicit: the Type I error bound includes an inflation term driven by estimation error. We'll emphasize that conditional estimation is a core statistical component.
>
> We also agree that high-dimensional C, A, and B make CI testing more challenging. In our framework, this manifests through harder conditional estimation and increased sensitivity to representation and kernel choice. In our synthetic neural data, while A and B are high-dimensional, the conditional dependence is "clear" enough that simple kernels work. We do not claim to solve high-dimensional CI testing in full generality, but our approach can be combined with representation learning to improve practical performance. (In the offline setting, see He et al., 2025.)
>
> To further evaluate this, we added experiments on the dSprites dataset, using low-dimensional learned features. Here $B$ is the full image, $C$ is a partial image patch, and $A$ is the object label, so the CI task tests whether $C$ captures all information about $A$ contained in $B$.
>
> Our method continues to outperform baselines. In the pretrained setting, where conditional mean estimates are relatively accurate, DAVT  inflated type I error (exceeding 0.09), while e-CRT maintains calibration but accumulates power relatively slowly. Our method achieves a better balance.
>
> Type I Error: dsprites dataset, pretrain mode
> |Series|200|400|800|1200|2000|
> |:-|:-|:-|:-|:-|:-|
> |SKCI|0.01±0.01|0.03±0.02|0.03±0.02|0.03±0.02|0.04±0.02|
> |DAVT|0.02±0.01|0.08±0.03|0.09±0.03|0.09±0.03|0.10±0.03|
> |e-CRT|0.01±0.01|0.01±0.01|0.03±0.02|0.04±0.02|0.04±0.02|
>
> Power: dsprites dataset, pretrain mode
> |Series|200|400|800|1200|2000|
> |:-|:-|:-|:-|:-|:-|
> |SKCI|0.80±0.04|1.00±0.00|1.00±0.00|1.00±0.00|1.00±0.00|
> |DAVT|0.47±0.05|1.00±0.00|1.00±0.00|1.00±0.00|1.00±0.00|
> |e-CRT|0.37±0.05|0.89±0.03|0.96±0.02|0.98±0.01|0.99±0.01|
>
> In the online setting, where conditional estimation is more challenging, both DAVT and e-CRT exhibit severe type I error inflation (approaching 1). In contrast, our method remains substantially more robust, though we note that significant inflation (up to ~0.24) remains due to the inherent difficulty of estimation in high dimensions.
>
> Type I Error: dsprites dataset, online mode
> | Series| 200 | 400 |800 | 1200| 2000|
> |:-|:-|:-|:-|:-|:-|
> | SKCI| 0.06±0.02 | 0.13±0.03 | 0.19±0.04 |0.23±0.04 | 0.24±0.04 |
> | DAVT| 0.98±0.01 | 1.00±0.00 | 1.00±0.00 |1.00±0.00 | 1.00±0.00 |
> | e-CRT| 0.77±0.04 | 1.00±0.00 |1.00±0.00 |1.00±0.00 | 1.00±0.00 |
>
> Power: dsprites dataset, online mode
> | Series| 200| 400 ... 2000|
> |:-|:-|:-|
> | SKCI| 0.99±0.01 | 1.00±0.00 |
> | DAVT| 1.00±0.00 | 1.00±0.00 |
> | e-CRT| 1.00±0.00 | 1.00±0.00 |
>
> ### Kernel Choice, hyperparameter selection, and algorithmic details
> In our current implementation, kernel choices follow standard practice in kernel-based CI testing, with bandwidth selected via standard heuristics or validation. Some details were deferred to Appendix B; we will integrate references to this better in the main text.
>
> In the revision, we will make the procedure more explicit by specifying kernel choices, bandwidth selection, hyperparameter tuning, and the conditional estimation step, as well as clarifying the round-by-round workflow. We will also expand the discussion of how these choices adapt in high-dimensional settings.
>
> ### Comparison with Model-X methods under approximate model-X
>
> We agree this comparison is important. We include such results in Appendix C.1 (as well as Figure 2b in the main body), and will highlight them more clearly. The overall picture is that when the Model-X assumption approximately holds, Model-X methods can perform well; our method remains competitive. When it is misspecified, their calibration can degrade significantly, while our method remains substantially more robust. We will make this comparison more explicit in the revised version.
>
> ---
>
> We sincerely thank the reviewer for the positive and encouraging evaluation. We are glad that the reviewer found the problem and our approach meaningful. We will incorporate the suggested improvements to further strengthen the paper.

---

> > ### Author Rebuttal · Reviewer_YLnf · 2026-04-02
> >
> > I appreciate the authors’ efforts to address the comments and questions. Given the limited time during the rebuttal period, I understand that it may not be possible to fully address all concerns.

---

> > > ### Author Response · Authors · 2026-04-05
> > >
> > > Thanks again for the thoughtful feedback and for carefully considering our responses.
> > >
> > > We noticed that you mentioned having follow-up questions. We would be very happy to address them—if there are any specific remaining points where clarification would help, please feel free to let us know. We will do our best to respond as concretely as possible.

---

### Official Review · Reviewer_HEJw · 2026-03-11

**Soundness:** 2
**Presentation:** 3
**Significance:** 2
**Originality:** 2
**Overall Recommendation:** 4
**Confidence:** 3

**Summary:**

The authors tackle a very challenging problem of conditional independence testing without the typical "Model-X" assumption. We are interesting in testing the null hypothesis $A\perp B \mid C $. The "Model-X" assumption assumption assumes that the $A\mid C$ distribution is known. The authors oppose this assumption as it is not realistic in practice. However, most works make this assumption, because without it the problem is "impossible". Shah & Peters (2020) formally established that achieving exact Type I error control is impossible without making further assumptions (like Model-X). Shah & Peters demonstrate that for any fixed sample size dataset, there exists at least a distribution under the null and alternative that are indistinguishable from each other. From the testing perspective, it is not possible to have strict Type I error and nontrivial power at the same time.

The authors sidestep this impossibility result by redefining the null in terms of a kerneled null hypothesis, having mapped the problem to an RKHS. Their empirical results are encouraging on simulated and real datasets, but I do have some concerns regarding the choices made in construction of their test martingale, elaborated below. For example, to control estimation errors, the authors actually used fixed-t bounds, and then apply a union bound across time, up to a maximum time T. The union bound over t is a little inelegant, and the maximum T renders it not truly anytime-valid. Also given that the data is batched, only a finite number of interim analyses are applied, and they are setting a maximum T, then why not do group sequential testing instead of anytime-valid?

**Compliance With Llm Reviewing Policy:**

Affirmed.

**Final Justification:**

The authors have provided an adequate treatment to a very hard problem. It seems, as other reviewers have noted, that some choices are a bit ad-hoc, and it's not clear to me whether these choices are optimal. Nevertheless, it is an interesting paper to a relevant problem, and I think it is a useful contribution, so I'll keep it as weak accept.

**Key Questions For Authors:**

The wealth martingale you use, $W_t = W_{t−1}(1 + \lambda_t V_t)$, requires $V_t$ to be bounded below by -1. You enforce this using $\max(\dots,-1)$ , but the mean of $\max(\dots,-1)$ is no longer zero under the null ; clipping it at -1 forces the mean up, so you need to subtract $\gamma_t$, setting $V_t = \max(V_t^{raw}-\gamma_t,-1)$, to recover the zero-mean property. Estimating this $\gamma_t$ parameter then requires an intensive wild bootstrap. This is a massive amount of algorithmic effort (clipping, recentering, bootstrapping) strictly to satisfy the $[-1,\infty)$ support requirement of this specific linear payoff. Why not sidestep this issue entirely and use a martingale valid for random variables in $(-\infty, \infty)$? I recognize that your current approach allows you to construct a finite-sample anytime validity bound (Theorem A.5) by bounding the Wasserstein distance to the Gaussian limit. However, since you heavily rely on asymptotic Gaussian arguments in Section 4 to calculate the shift regardless, why not just approximate your test statistic as a Wiener process and use an asymptotic Gaussian confidence sequence?Since you already have an estimator $\hat{\sigma}_t^2$ for the variance of $V_t^{raw}$, you could construct an asymptotic e-process using a Gaussian mixture martingale. For example (where $t$ represents the accumulated steps):
$$e_t = \sqrt{\frac{\lambda}{\lambda+t}}e^{\frac{1}{2}\frac{t}{t+\lambda}\left(\frac{V_t^{raw}}{\hat{\sigma}_t}\right)^2}$$Have you experimented with this approach to bypass the computational bottleneck of the wild bootstrap and the truncation bias entirely?

What if you didn't bother to control the estimation error, and just used the raw $V_t^{\raw}$ instead of $U_t$? I'd be curious to learn if the Type I error is really that bad.

**Limitations:**

--

**Strengths And Weaknesses:**

The H_0 subscript on line 89 seems formatted incorrectly.

Strengths:
- Biggest strength is totally removing the restrictive "Model-X" framework. The authors demonstrate that the Type-I error of the DAVT model-X competitors breaks down when the conditional distribution p(A|C) needs to be estimated.

Originality
- Many of the idea seem to be present in Podkopae et al "Sequential Kernelized Independence Testing"

Weaknesses:
- ECRT method still appears to give good type I error calibration, even though it is a model-X method with estimated distribution.

- The truncation and shift seems a little inelegant in the presence of alternatives (see question to authors)

- Wild bootstrap at each interim analyses is computational burdensome.

- Not true "Finite Sample Anytime Validity": A significant theoretical limitation of the proposed method is that its "anytime validity" guarantees actually depend on a pre-specified, finite maximum time horizon $T$. In Lemma 4.2, the upper bound on the conditional drift, $U_t$, explicitly includes a bootstrap noise term that scales with $\sqrt{\log(2T/\tau)}$. As detailed in Appendix A.4, this dependency exists because the authors must apply a standard union bound across all time steps up to $T$ (specifically, replacing the failure probability $\tau$ with $\tau/T$) to ensure the bootstrap variance concentration bounds hold simultaneously across the entire sequence. This union-bound across T is a "poor-man's" anytime-valid approach. Consequently, this does not yield a true infinite-horizon anytime-valid test ($T \rightarrow \infty$), which is the standard in the testing-by-betting literature. If the test is run indefinitely, the drift bound $U_t$ mathematically diverges, and the finite-sample validity guarantee (Theorem A.5) collapses. Practitioners must artificially cap their experiment at a pre-determined $T$, which undermines the core "optional stopping" flexibility that the betting paradigm is meant to provide.

On the other hand, this is all in the spirit of attempting to bound the approximation error from their estimated terms. If you want finite sample anytime-validity, you would need a time-uniform bound on this approximation error also. The authors are currently doing this by applying a fixed-t error bound, and union bounding it from t=1,..., T. The attempt is admirable, and other papers don't even go as far as this. But, again, why not just use the asymptotic Gaussian confidence sequence?

---

> ### Author Rebuttal · Authors · 2026-03-30
>
> Thanks for the detailed and insightful feedback. We'll clarify in revision!
> ### On anytime validity
> We agree that our method is not infinite-horizon anytime valid.
>
> In the Pretrained mode, fixed conditional estimates imply that any consistent test will eventually detect misspecification and reject, so exact anytime validity is impossible. In the Online mode, one could assume convergence rates of the estimates to obtain asymptotic or (unnecessarily) conservative finite-sample guarantees, but such assumptions are typically unverifiable in practice.
>
> Our approach instead asks: if we simply run the algorithm, how bad can it be? While one might hope that summable conditional errors over time would yield an anytime guarantee, the $\log T$ term in our bound precludes this. Our result instead gives an explicit bound on how wrong the test could be over a given runtime.
>
> Group sequential testing for fixed-horizon tests is indeed natural. Our goal, however, is a horizon-free test algorithm, where only the analysis—not the procedure itself—depends on a fixed T.  We refer to our response to Reviewer Lzgo for additional long-horizon experiments.
> ### On Gaussian / Wiener alternatives
>
> We implemented and evaluated two methods below; neither is as calibrated as our proposal.
>
> Wiener-process approaches require the cumulative statistic to have (approximately) independent increments, but ours depend on the kernels and CME estimates from the full history.
>
> Exponential Gaussian e-processes only require conditional asymptotic normality. Yet outside Model-X, the conditional mean is not exactly zero and its variance must be estimated, introducing many of the same challenges as our proposal.
>
> Type I Error: CI hardness data, d=1, online mode
> |Process|200|400|800|1200|2000|
> |:-|:-|:-|:-|:-|:-|
> |$V_t$ (ours)|0.00±0.00|0.01±0.01|0.02±0.01|0.03±0.02|0.03±0.02|
> |Wiener process|0.38±0.05|0.64±0.05|0.79±0.04|0.92±0.03|0.97±0.02|
> |Exp. Gaussian|0.00±0.00|0.03±0.02|0.08±0.03|0.11±0.03|0.14±0.03|
>
> Power: CI hardness data, d=1, online mode
> |Process|200|400 ... 2000|
> |:-|:-|:-|
> |$V_t$ (ours)|0.74±0.04|1.00±0.00|
> |Wiener process|0.93±0.03|1.00±0.00|
> |Exp. Gaussian|0.49±0.05|1.00±0.00|
> ### Truncation-shift construction
>
> This construction arises because $V_t^{\mathrm{raw}}$ does not yield a valid betting process under our payoff. Truncation enforces the required constraint, and shifting corrects the bias introduced by truncation.
>
> The shift $\gamma$ also provides a robustness benefit. Our variance estimate is based on ${E}[Z^2]$, implicitly assuming ${E}[Z]=0$. When conditionals are misspecified (${E}[Z]\neq 0$), the variance is overestimated; since $\gamma$ increases with variance, this induces a conservative shift that helps offset positive drift from regression error.
>
> In light of Lemma 4.2: the leading asymptotic gap is of order $\rho/\sqrt{b}$. Increasing $\gamma$ at a comparable scale partially compensates for this term. This is reflected in ablations below: $V_t^{\mathrm{raw}}$ inflates Type I error, while increasing $\gamma$ improves calibration at a cost of power. We will clarify this dual role of $\gamma$ in the revision.
>
> Type I Error: CI hardness data, d=3, shared coordinate, online mode
> |Vt type|200|400|800|1000 ... 2000|
> |:-|:-|:-|:-|:-|
> |$V_t$ (default)|0.03±0.02|0.05±0.02|0.07±0.03|0.08±0.03|
> |$V_t$ (increase $\hat\gamma_t$ by 0.1)|0.02±0.01|0.02±0.01|0.02±0.01|0.02±0.01|
> |$V_t^{\mathrm{raw}}$|0.13±0.04|0.13±0.04|0.15±0.05|0.15±0.05|
>
> Power:  CI hardness data, d=3, shared coordinate, online mode
> |Vt type|200|400 ... 2000|
> |:-|:-|:-|
> |$V_t$ (default)|0.83±0.04|1.00±0.00|
> |$V_t$ (increase $\hat \gamma_t$ by 0.1)|0.72±0.05|1.00±0.00|1.00±0.00|
> |$V_t^{\mathrm{raw}}$|0.89±0.03|1.00±0.00|
> ### Cost of bootstrap
> Thanks for bringing up this important point. The bootstrap can actually be removed. Given the variance estimate, we can use binary search on the equation $f(\gamma;0,\hat\sigma)=0$ in Appendix A.1 to obtain $\hat\gamma_t$, reducing runtime substantially (e.g., 13ms to 0.4ms).
> ### ECRT performance
> ECRT does indeed have reasonable type I control in some settings (but not always, as in Figure 2b). In the non-Model-X regime, however, when ECRT has good validity, it often has quite poor power, as in Figures 2-4. Our method generally maintains a better validity–power trade-off.
> ### On originality
> The key distinction from the prior work you mention lies in the regime. In unconditional independence testing, there is no need to estimate conditional objects, and exact martingale constructions are readily available. In contrast, our setting requires estimating conditional quantities online, which introduces key statistical and algorithmic challenges.
>
> ---
>
> Thanks again for the thorough and thoughtful comments; they've led to meaningful improvements in both the implementation and presentation. We hope our revisions and clarifications help, and that you'd consider updating the evaluation accordingly.

---

> > ### Author Rebuttal · Reviewer_HEJw · 2026-04-01
> >
> > Thank you. While I agree with reviewer 6iwf that many of the methodological choices seem quite ad-hoc, I will update my recommendation to weak accept.

---

### Official Review · Reviewer_6iwf · 2026-03-12

**Soundness:** 2
**Presentation:** 2
**Significance:** 2
**Originality:** 2
**Overall Recommendation:** 3
**Confidence:** 3

**Summary:**

This paper presents a new way to do anytime-valid conditional independence testing in an online (i.i.d.) setting in a way that updates online to increase power and maintain type I error control when the conditional distributions are unknown (and thus must be estimated).

**Compliance With Llm Reviewing Policy:**

Affirmed.

**Final Justification:**

As described in my rebuttal acknowledgement, the rebuttal raised my score from a 2 to a 3 by highlighting, and saying they will reframe the paper around, a different contribution than the original methodological contribution the paper was framed around. I did not raise my score beyond a 3 because I think this reframing is a very large one so I can't know how it will go, and I still feel there are unaddressed weaknesses in the methodological contribution, particularly its claims of robustness against violations of the model-X assumption.

**Key Questions For Authors:**

I’m afraid as I don’t find the theory very compelling, answering the following questions is unlikely to change my evaluation.

Is the “main validity result” in 4.3 proved? It is not completely immediate to me: I’m not sure what it means that \tilde{M}_t is a supermartingale conditioned on an event that depends on the data.

Lemma 4.2’s bound holds with probability 1-\tau at each t, or uniformly over all t?

**Limitations:**

Yes.

**Strengths And Weaknesses:**

Careful consideration of the payoff construction in betting-based anytime-valid conditional independence testing seems worthwhile. They present quite a few empirical results that demonstrate superior performance compared to two existing methods by a large margin (though I am not an expert in this particular sub-area of anytime valid conditional independence testing, so I am not sure whether this is a fair or state-of-the-art comparison). However, they also show small type I error violations of the proposed method in some cases (which I was surprised by since anytime valid methods are typically extremely conservative), and it’s hard to consider a handful of empirical results compelling for (approximate) validity on their own (I will return to the theory in a few lines). Methodologically, I feel this paper’s choices are more heuristic than natural or justified in a principled way (e.g., the shift and truncate approach, the epsilon denominator to avoid division by zero, and the choice of Gaussian kernel all seem rather ad hoc), and their theory (which ultimately only gives a bound on type I error inflation that is hard to interpret/quantify and appears to decay exponentially in time) does not particularly support or explain the empirics or the pitch made in the abstract and introduction.

So one concern is about soundness/presentation: I don’t think the paper after section 1 lives up to the promises made in the abstract and last two paragraphs of section 1. Another concern is originality/significance: this paper is addressing a problem that has been addressed before by other works, and it is unclear what major contribution it is making to the literature. Perhaps the authors are onto something in terms of power with their payoff construction, but I ultimately do not find their validity claims (which seem central to their abstract and intro) convincing.

---

> ### Author Rebuttal · Authors · 2026-03-30
>
> We thank the reviewer for the careful feedback. We agree that the key issue is how to understand validity once conditional distributions must be estimated, and we appreciate the opportunity to clarify both our claims and contributions.
>
> ###  Clarifying our validity claim and positioning
> We agree that the current presentation overstates “anytime validity”. Without additional assumptions, no method with nontrivial power can be exactly valid even at a single fixed time point (Shah & Peters et al., 2020). Our method therefore does not provide exact finite-sample anytime validity. Instead, under estimated conditionals, Theorem 4.3 shows that the rejection probability is controlled up to an explicit multiplicative factor depending on accumulated drift terms. We will revise the abstract and introduction to clarify that our goal is a controlled relaxation of validity, rather than an exact guarantee.
>
> Importantly, our contribution is not only to acknowledge that validity degrades, but to quantify _how_ it degrades. The drift decomposition separates three sources: finite-sample approximation, regression bias, and variance estimation noise (Section 4.2).
>
> We agree that the current version does not connect this bound to empirical behavior as deeply as it could. In revision, we will strengthen this connection in two ways. First, we will explicitly report proxies for these terms and show how they correlate with observed Type I error. Second, we will highlight existing empirical evidence already included in the appendix. In Figure 9, increasing the batch size reduces the Type I error, consistent with the $O(1/\sqrt{b})$ term in the drift bound. Moreover, across all experiments (Figures 1–8), the Oracle and Pretrained modes consistently achieve lower Type I error than the fully Online setting. This pattern aligns with the regression bias term: as conditional estimation improves, the drift decreases and validity is restored. We will make this connection explicit to show that the theory captures not only qualitative behavior but also the correct trends observed in practice.
>
> Your comments also helped us identify a typo: in Lemma 4.2, $U_t$ depends on estimation errors of the conditional means, not the means themselves. We will correct this.
>
> ### Clarifying originality
> Our contribution is not a new CI statistic per se, but a new regime: sequential CI testing without Model-X assumptions, together with a construction that remains stable when conditionals are estimated. Prior betting-based methods rely on exact conditionals, losing validity (without any attempt at characterizing by how much) when this assumption fails. Our method provides a concrete procedure whose deviation from validity is explicitly controlled, while empirically maintaining strong power in settings where existing methods either become overly conservative or break down.
>
> ### On methodological choices
>
> Our design choices, while unavoidably a little ad hoc, do follow a single principle: constructing a payoff that quantifies distributional discrepancies while enforcing the constraints required for valid betting under misspecification. The normalization controls the scale of wealth update to retain power, the truncation enforces nonnegativity, and the shift provides a data-driven correction under an unknown null. We will revise Section 3 to present it as a unified construction (a robustified e-process) rather than as separate design elements.
> Alternative constructions, see response to Reviewer HEJw, are empirically less well-calibrated than our approach.
>
> For the kernel choices, we follow standard practice in kernel-based CI testing for comparability (e.g., Pogodin et al., 2024; He et al., 2025), as described in Appendix B. The framework itself is not tied to a specific kernel and can naturally incorporate alternative representations.
>
> ### Clarifying Theorem 4.3 (“main validity result”) and Lemma 4.2 (uniform vs pointwise)
> Yes, the result is formally proved in Appendix A (Theorem A.5). The argument conditions on an event E under which the variance concentration bounds hold uniformly over $t \le T$ rather than pointwisely. On this event, the compensated process is a nonnegative martingale under $H_0$.
>
> We agree this is not sufficiently clear in the main text and will revise the presentation.
>
> ---
>
> In summary, we will revise the paper to align the claims with the theory, clarify the design principle, and more directly connect theoretical quantities to empirical behavior. We believe this better reflects the contribution: extending betting-based testing to a realistic regime where exact validity is impossible, but where validity degradation can be quantified and controlled in practice.
>
> We sincerely thank the reviewer again for the thoughtful comments. We hope the reviewer finds the revisions and clarifications helpful and that we have addressed their concerns. If anything remains unclear, we would be happy to discuss further and provide additional clarification.

---

> > ### Author Rebuttal · Reviewer_6iwf · 2026-04-02
> >
> > I thank the authors for clarifying their contribution, which I now understand to be both (and perhaps primarily) about understanding the consequences of relaxing the model-X assumption as well as a method to handle this setting. I do think the authors' promised reframing to better highlight the former contribution will improve the paper, prompting me to raise my score. However, I want to point out that this is a major reframing--I find the current presentation is very focused on the method--and hence I'm not sure it's fair to recommend acceptance for such a hypothetical revised paper base only on high-level descriptions in the authors' rebuttals of how it will be revised.
> >
> > Furthermore, I still feel the rebuttal overstates the validity of the proposed procedure. The rebuttal states that the proposed method's "deviation from validity is explicitly controlled", which suggests that a user has some way to control its deviation from validity, which I don't think the paper makes a case for--indeed as the authors acknowledge in their rebuttal's reference to Shah and Peters (2020), no method can really control its deviation from validity without some assumptions. The rebuttal also states that the method "remains stable when conditionals are estimated", but as I mentioned in my original review, I don't see sufficient evidence for this claim to hold in general. If anything, the theory's exponential type I error degradation suggests strong instability over any reasonable time horizon.
> >
> > Taken together, I remain overall against acceptance, but less so than before, so I will raise my score to a 3.

---

> > > ### Author Response · Authors · 2026-04-05
> > >
> > > Thanks for the careful follow-up and for raising the score. We appreciate the opportunity to clarify and address the reviewer’s remaining concern.
> > >
> > > We appreciate the concern regarding the reframing. Our intent is not to change the paper’s technical content or contributions, but to better align the presentation with what is already established in the current submission: a testing method together with a theoretical analysis of how validity degrades when the Model-X assumption is relaxed. The planned revision is therefore limited in scope. In particular, we would not alter the core method, theorem statements, or experimental setup. Rather, we would clarify the claims in the abstract and introduction, correct the wording around "anytime validity", and more explicitly connect the existing drift decomposition and appendix evidence to the observed Type I error behavior.
> > >
> > > On the validity claim, we agree that our previous wording may have suggested a stronger claim than intended. We do not mean that the user can "control" deviation from validity in a strict sense. Rather, Theorem 4.3 provides an explicit upper bound on how the rejection probability can exceed the nominal level as a function of estimation error and batch size. This is a characterization of how validity may degrade under estimated conditionals, not a guarantee that can be tuned to achieve exact validity without additional assumptions. We will revise the wording accordingly.
> > >
> > > Similarly, our intended claim regarding Type I error behavior is more limited. Empirically, under estimated conditionals, our method shows substantially smaller Type I error inflation than the baselines. Moreover, in the new longer-horizon experiments, we do not observe continued growth of Type I error over time after an initial phase; instead, the error appears to level off. For example, on the CI-hardness data in online mode with batch size 20, the observed Type I error for SKCI remains essentially flat from $t=100$ to $t=300$: for $d=1$, it stays at $0.02 \pm 0.01$, and for $d=3$ with a shared coordinate, it stays at $0.09 \pm 0.03$.
> > > We will revise the text to state this empirical finding directly, rather than suggesting a general stability guarantee.
> > >
> > > We further note that the theory characterizes how estimation error and batch size can affect Type I error through a worst-case upper bound, but this bound is not intended to tightly capture exact finite-sample behavior. Consistent with this, the Type I error growth permitted by the bound is not what we observe empirically.
> > > We will clarify that the analysis provides a conservative upper bound on possible degradation, while the experiments illustrate the practical behavior of the procedure across a broad range of settings we study.
> > >
> > > Overall, we will revise the paper to (i) align the presentation with the actual scope of the results, and (ii) make all claims about validity and Type I error behavior more precise and conservative. We appreciate the reviewer’s feedback in helping us sharpen these points.

---

### Decision · Program_Chairs · 2026-04-30

**Decision:**

Accept (regular)

**Comment:**

This work addresses a challenging question: performing conditional independence test without the popular "model X" assumption. Without this assumption, it is impossible in general to provide meaningful type 1 error guarantees. This work bypasses this impossibility by relaxing the null hypothesis using the kernel method. Reviewers appreciate the originality in tackling this challenging question and believe that the proposed solution is promising. There is strong concern regarding the paper presentation (over-statement, lack of clarity, insufficient justification) that has lead to low initial scores. Reviewers raised their scores after the authors promised significant revision and additional experimental results in the rebuttal. I trust the authors to implement these revisions and include promised experimental results in the camera-ready version. I recommend acceptance.